# Enhanced warming of European mountain permafrost in the early 21st century

Jeannette Noetzli [1,2] ✉, Ketil Isaksen [3], Jamie Barnett [4], Hanne H. Christiansen [5], Reynald Delaloye [6], Bernd Etzelmüller [7], Daniel Farinotti [8,9], Thomas Gallemann [10], Mauro Guglielmin [11], Christian Hauck [6], Christin Hilbich [6], Martin Hoelzle [6], Christophe Lambiel [12], Florence Magnin [13], Marc Oliva [14], Luca Paro [15], Paolo Pogliotti [16], Claudia Riedl [17], Philippe Schoeneich [18], Mauro Valt [19], Andreas Vieli [20] & Marcia Phillips [1,2]

Mountain permafrost, constituting 30% of the global permafrost area, is sensitive to climate change and strongly impacts mountain ecosystems and communities. This study examines 21st century permafrost warming in European mountains using decadal ground temperature data from sixty-four boreholes in the Alps, Scandinavia, Iceland, Sierra Nevada and Svalbard. During 2013–2022, warming rates at 10 metres depth exceed 1 °C dec⁻¹ in cases, generally surpassing previous estimates because of accelerated warming and the use of a comprehensive data set. Substantial permafrost warming occurred at cold and ice-poor bedrock sites at high elevations and latitudes, at rates comparable to surface air temperature increase. In contrast, latent heat effects in ice-rich ground near 0 °C reduce warming rates and mask important changes of mountain permafrost substrates. The warming patterns observed are consistent across all sites, depths and time periods. For the coming decades, the propagation of permafrost warming to greater depths is largely predetermined already.

While the retreat of glaciers in the 21st century has been widely recognised[1,2], permafrost changes in cold regions worldwide are far less visible despite being similarly important[3–12]. Permafrost is a key component of the cryosphere and is defined thermally and temporally as ground with a maximum temperature of 0 °C over at least two consecutive years. Mountain permafrost constitutes 30% of the global permafrost area[13,14] and spans low to high latitudes in both hemispheres and all continents. Complex topography leads to large

[1]WSL Institute for Snow and Avalanche Research SLF, Davos Dorf, Switzerland. [2]Climate Change, Extremes and Natural Hazards in Alpine Regions Research Centre CERC, Davos Dorf, Switzerland. [3]Norwegian Meteorological Institute, Oslo, Norway. [4]Department of Geological Sciences, Stockholm University, Stockholm, Sweden. [5]Arctic Geophysics Department, University Centre in Svalbard, Longyearbyen, Norway. [6]Department of Geosciences, University of Fribourg, Fribourg, Switzerland. [7]Department of Geosciences, University of Oslo, Oslo, Norway. [8]Laboratory of Hydraulics, Hydrology and Glaciology (VAW), ETH Zurich, Zurich, Switzerland. [9]Swiss Federal Institute for Forest, Snow and Landscape Research WSL, Birmensdorf, Switzerland. [10]Bavarian Environment Agency, Augsburg, Germany. [11]Department of Theoretical and Applied Science, Insubria University, Varese, Italy. [12]Institute of Earth Surface Dynamics, University of Lausanne, Lausanne, Switzerland. [13]Laboratoire EDYTEM, CNRS/Université Savoie Mont-Blanc, Le Bourget-du-Lac, France. [14]Department of Geography, Universitat de Barcelona, Barcelona, Spain. [15]Environmental Protection Agency of Piedmont, Turin, Italy. [16]Environmental Protection Agency of Valle d'Aosta, Saint Christophe, Italy. [17]GeoSphere Austria, Salzburg, Austria. [18]PACTE, Institut d'Urbanisme et de Géographie Alpine, Université Grenoble Alpes, Grenobles, France. [19]Environmental Protection Agency of Veneto, Centro Valanghe di Arabba, Arabba, Italy. [20]Department of Geography, University of Zurich, Zurich, Switzerland. ✉e-mail: jeannette.noetzli@slf.ch

environmental gradients in these mountain areas and controls the permafrost distribution e.g., refs. [15],[16]. Mountain permafrost is highly sensitive to the pronounced atmospheric warming observed at high elevations[17]. This permafrost warming and degradation has implications for the stability of steep, perennially frozen mountain slopes[18–24] with potentially far-reaching impacts and risks to both human safety[25–27] and infrastructure[28], as well as effects on ecosystems[29–32] and hydrological processes[33–36].

The assessment of permafrost changes relies on long-term records of ground temperatures measured in boreholes, which are the direct thermal observations defined as one of the products of the Essential Climate Variable (ECV) permafrost[37–39]. Temperatures near the ground surface follow the fluctuations in atmospheric conditions. These variations are increasingly attenuated and delayed with depth. At the Depth of Zero Annual Amplitude (DZAA) and below, seasonal temperature changes become negligible and variations result from long-term climate-related changes. Near the DZAA, Biskaborn et al.[3] reported global permafrost warming of 0.29 °C dec$^{-1}$ and of 0.19 °C dec$^{-1}$ for mountain regions during the period 2007–2016. The study included 28 mountain permafrost sites, of which a large part is located in Europe. This is where the most densely populated permafrost landscapes are found and where – motivated by the growth of tourism and infrastructure development – systematic research on mountain permafrost commenced in the late 1970s[40–42] for a review see ref. [43]. Climate-related European permafrost monitoring was initiated in the late 1990s with a borehole transect from high Arctic Svalbard to the Mediterranean Sierra Nevada Massif[7,44]. Although several national and regional permafrost observation programs have been established in recent decades[45–51] and activities in the framework of the Global Terrestrial Network for Permafrost (GTN-P) have increased[37], there is no pan-European coordination and synthesis of mountain permafrost temperature data to date. Regional and local studies point to relevant differences in permafrost warming rates due to the high variability in topographic and (sub-)surface conditions in mountain regions[7–9,11,52–54]. However, they are typically restricted to single sites or regions and are not directly comparable.

In this study, we present warming patterns of mountain permafrost across Europe up to the year 2022. We compiled a consistent data set of 64 ground temperature time series measured for at least one decade in or near permafrost areas. Data are obtained in boreholes extending at least 10 m in depth and the sites cover a wide range of landforms, elevations and latitudinal zones from high Arctic Svalbard and Scandinavia to Iceland, the Alps and the Sierra Nevada. To assess the spatial and temporal variability of warming patterns, we derive warming rates based on annual mean values of ground temperatures (MGT$_{1yr}$) for recent 10- and 20-year periods (2013–2022 and 2003–2022) at 5, 10 and 20 m depth. We find evidence for permafrost warming for all considered regions, depths, and time periods. The warming rates identified on the basis of the comprehensive and current data set exceed previous estimates for mountain permafrost[3] and match the levels observed in the permafrost of the Arctic hotspot regions[4]. The warming patterns are primarily dependent on the depth of observation and the exchange of latent heat for phase change. In ice-bearing ground, the latter can significantly reduce warming rates when temperatures approach 0 °C thereby masking important changes such as increasing water content or ground ice melt. Warming rates are highest at cold permafrost sites with low ground ice content, where latent heat effects are largely absent. Further, warming rates in the uppermost 10 m are higher than at greater depths, where temperatures react with delay. These patterns prove consistent across all depths and time periods considered and surpass regional differences.

## Results

### Measured ground temperatures in European mountains

Our data compilation provides decadal time series of ground temperatures measured in 64 boreholes at 39 sites in European mountain permafrost regions between 37 and 78° N and at elevations between 275 and 3800 m asl., and in nine countries[55] (Fig. 1, Table 1, Supplementary Figs. S1–S2, for detailed data origin see Supplementary Table S1). Two boreholes are situated on Svalbard, 13 in Scandinavia (Norway, Sweden), four on Iceland, 15 in the Eastern and Central Alps (Austria, Germany, Switzerland, Italy), 29 in the Western Alps (Switzerland, Italy, France) and one in the Spanish Sierra Nevada. Data from additional sites in these regions exist, but they do not meet the minimum length or observation depth. No data are available yet from other European mountain regions such as the Pyrenees. Sites are in landforms typically found in mountain permafrost such as talus or bedrock slopes, rock crests, plateau blockfields, rock glaciers, or moraines. Nine boreholes are not in permafrost from the start of observations but are located close by. They are included in the analyses for comparison. The longest time series is measured in rock glacier Murtèl-Corvatsch and covers 35 years. Four time series from the Western and Central Alps only cover a decade.

Permafrost temperatures recorded in European mountains range from cold permafrost with a minimum MGT$_{1yr}$ of −6.2 °C at 10 m depth to very warm permafrost with MGT$_{1yr}$ at or close to 0 °C (Fig. 2). MGT$_{1yr}$ below −4 °C are measured at the highest latitudes and elevations in High Arctic Svalbard (Breinosa, Janssonhaugen) or in North-facing bedrock slopes at elevations above 3500 m asl. in the Western Alps (Jungfrau Ridge, Aiguille du Midi). MGT$_{1yr}$ below −12 °C were measured near the surface on northern slopes in the Western Alps at elevations well above 4000 m asl.[52], indicating very cold permafrost conditions. However, long-term observations at depth from such locations are not available to date. Permafrost with temperatures close to 0 °C is found in all European mountain regions, mainly in rock glaciers and unconsolidated sediments at elevations around 2500 m asl. in the Alps, or in blockfields and debris at elevations around 1500 m asl. in Scandinavia (e.g., at Bellecombes, Gentianes, Les Attelas, Dovrefjell, Juvvasshoe, or Tron). Permafrost temperatures measured in the Central and Eastern Alps are not as low as in the Western Alps. This is due to the lower elevations in the former regions and the lack of observations >3000 m asl., and less due to climatic differences. MGT$_{1yr}$ at the non-permafrost sites attain >3 °C (Veleta Peak and the south-exposed rock face at Gemsstock).

42 of the 57 time series (74%) with available MGT$_{1yr}$ data at 10 m depth for one of the past three years (2020–2022) reached their maximum value in these 3 years (Fig. 2). Similarly, 27 time series out of 36 (75%) reached their maximum MGT$_{1yr}$ at 20 m depth between 2020 and 2022 (Supplementary Fig. S3). The two sites on Svalbard (the time series with the lowest temperatures in Fig. 2a) experienced their maximum before 2020 due to the extremely warm years 2016–2018[56]. MGT$_{1yr}$ at 10 m depth increased to above 0 °C during the measurement period for 10 time series (17%), i.e. permafrost is no longer present at these sites and this depth.

### Permafrost warming rates

Temperature change rates at 10 m depth for the period 2013–2022 (or for the latest decade if time series extend at least to the year 2018, which is the case for 9 time series) range from −0.10 to 1.77 °C dec$^{-1}$ (Table 1, $n = 59$, 58% of $p$-values < 0.05). The mean warming rate for all sites where permafrost is present at this depth for at least part of the decade is 0.41 °C dec$^{-1}$ ($n = 50$, median 0.31 °C dec$^{-1}$, standard error of 0.05 °C dec$^{-1}$, 60% of $p$-values < 0.05). For 42% of the permafrost time series, the warming rate is higher than 0.5 °C dec$^{-1}$ ($n = 21$, 86% of $p$-values < 0.05). The highest 20% of the warming rates are above 0.7 °C dec$^{-1}$ and are typically measured in bedrock slopes or where

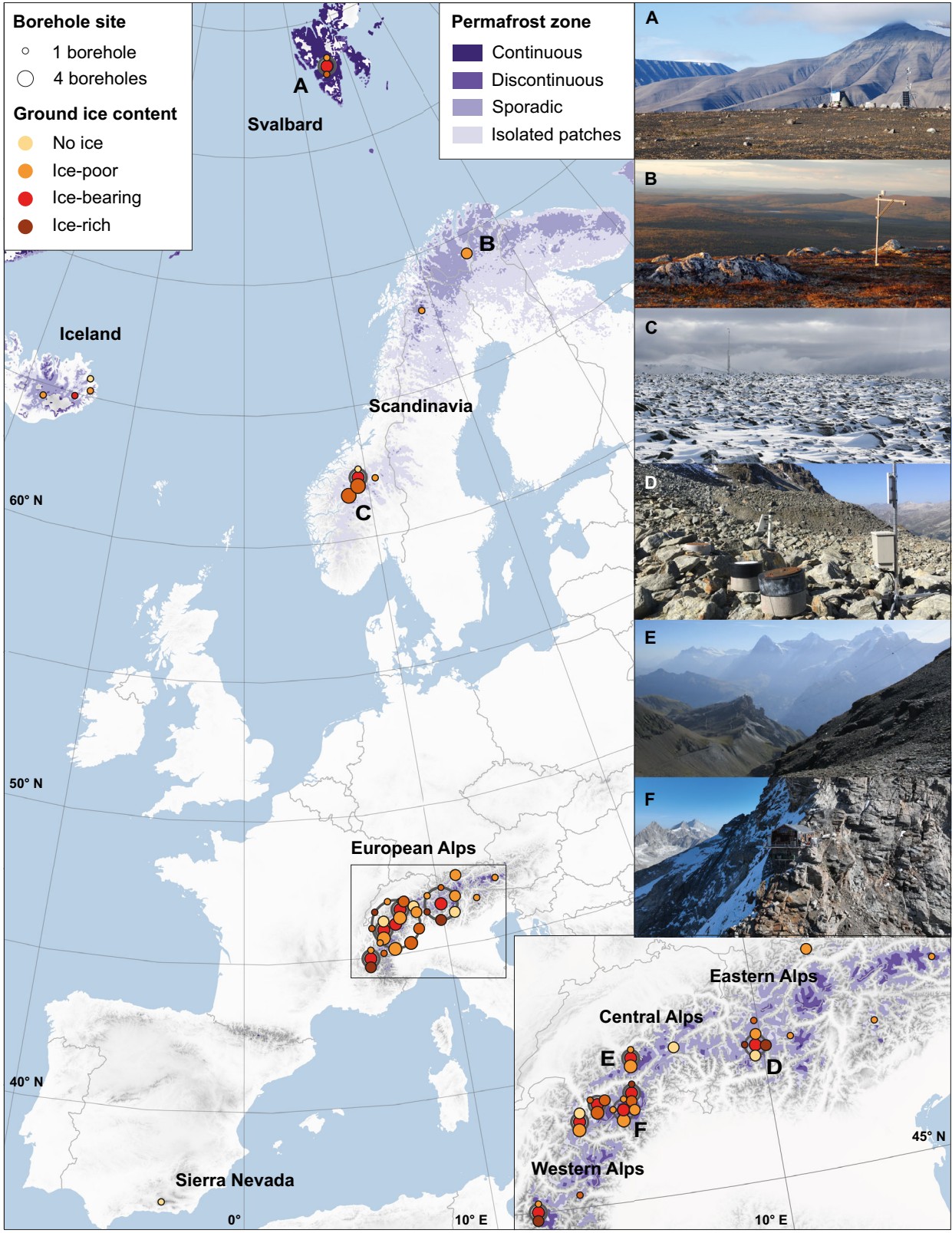

**Fig. 1 | Location of the permafrost observation sites in European mountain regions used in this study.** The 64 boreholes in European mountain regions with long permafrost temperature time series are obtained at 39 sites between Svalbard and the Sierra Nevada. A zoom on the Alpine Arc is given on the lower right. Each site is shown as one point. The size indicates the number of boreholes and the colour the general ground ice content (cf. Table 1). Overlapping points were offset for visibility (indicated by grey circles). On the upper right, photographs of several sites are shown to highlight the variable site characteristics. Data sources: permafrost zones are based on Obu et al.[14,88], background topography is from the GMTED2010[89], and administrative boundaries are from @EuroGraphics and the Food and Agriculture Organization of the United Nations (UN-FAO)[90]. Photographs show the sites Janssonhaugen (**A**), Iskoras (**B**), Dovrefjell (**C**), Murtél-Corvatsch (**D**), Schilthorn (**E**), and Capanna Carrel (**F**) and were taken by K. Isaksen (**A**–**C**), J. Noetzli (**D**), C. Hilbich (**E**), and P. Pogliotti (**F**).

**Table 1 | Boreholes in European mountains in permafrost areas and 2013–2022 warming rates at 10 m depth**

| Borehole name | Region | Elev. (m asl.) | Ground ice content | Surface cover | MGT$_{10yr}$ (min/max MGT$_{1yr}$) (°C) | Permafrost conditions | Warming rate (°C dec$^{-1}$) (n/R$^2$/p) |
|---|---|---|---|---|---|---|---|
| Aiguille du Midi NE | W Alps | 3745 | ice-poor | bedrock | −3.2 (−3.6/−2.8) | cold | 0.98 (9/0.93/0.00) |
| Aiguille du Midi NW | W Alps | 3738 | ice-poor | bedrock | −4 (−4.5/−3.6) | cold | 0.85 (10/0.88/0.00) |
| Aiguille du Midi S | W Alps | 3753 | ice-poor | bedrock | −1.1 (−1.5/−0.9) | warm | 0.82 (6/0.89/0.01) |
| Bellecombes amont | W Alps | 2725 | ice-rich | debris | −0.1 (−0.1/−0.1)* | near-zero | 0.00 (8/0.53/0.11) |
| Bellecombes aval | W Alps | 2710 | ice-rich | debris | −0.2 (−0.3/−0.2)* | near-zero | −0.07 (6/0.78/0.02) |
| Breinosa | Svalbard | 677 | ice-bearing | blockfield | −5.4 (−5.7/−5.0) | cold | 0.58 (8/0.37/0.11) |
| Capanna Carrel | W Alps | 3850 | ice-poor | bedrock | −2.4 (−2.7/−2.1) | cold | 0.73 (8/0.85/0.00) |
| Cime Bianche | W Alps | 3094 | ice-poor | bedrock | −0.8 (−1.0/−0.6) | warm | 0.39 (10/0.73/0.00) |
| Dovrefjell DB1 | Scand. S | 1505 | ice-bearing | debris | −0.1 (−0.2/−0.1)* | near-zero | −0.01 (5/0.03/0.79) |
| Dovrefjell DB2 | Scand. S | 1481 | ice-bearing | debris | −0.1 (−0.1/0) | near-zero | 0.08 (10/0.54/0.02) |
| Dovrefjell DB3 | Scand. S | 1477 | no ice | debris | 0.6 (0.3/0.8) | no | 0.55 (8/0.70/0.01) |
| Dovrefjell DB5 | Scand. S | 1458 | no ice | debris | 1.0 (0.6/1.4) | no | 0.80 (7/0.81/0.01) |
| Dovrefjell DB6 | Scand. S | 1402 | ice-bearing | debris | 0.1 (−0.3/1.1) | no | 1.77 (7/0.84/0.00) |
| Gagnheidhi | Iceland | 931 | ice-poor | debris | −0.2 (−0.4/−0.2)* | near-zero | 0.25 (8/0.82/0.01) |
| Gemsstock N | E Alps | 2940 | no ice | bedrock | 0.7 (0.5/1.0) | no | 0.34 (6/0.55/0.09) |
| Gemsstock S | E Alps | 2940 | no ice | bedrock | 2.6 (2.1/3.1) | no | 0.97 (6/0.53/0.10) |
| Gentianes 2002 | W Alps | 2888 | ice-bearing | debris | −0.2 (−0.4/−0.2) | near-zero | 0.19 (10/0.83/0.00) |
| Graechen BH1 | W Alps | 2451 | ice-bearing | debris | 2.1 (1.7/2.4) | no | 0.70 (10/0.82/0.00) |
| Graechen BH2 | W Alps | 2449 | ice-bearing | debris | 0.3 (0.1/0.7) | no | 0.80 (10/0.87/0.00) |
| Grapillon low | W Alps | 3000 | no ice | bedrock | 1.3 (1.0/1.5) | no | 0.63 (8/0.68/0.01) |
| Grapillon up | W Alps | 3100 | no ice | bedrock | 0.7 (0.4/0.9) | no | 0.17 (8/0.06/0.56) |
| Hágöngur | Iceland | 899 | ice-poor | debris | 0.1 (0.0/0.1)* | no | 0.08 (8/0.39/0.13) |
| Iskoras BH1 | Scand. N | 585 | ice-poor | bedrock | 0.6 (0.1/0.9)* | no | 0.75 (5/0.72/0.15) |
| Iskoras BH2 | Scand. N | 591 | ice-poor | debris | 0.4 (0.1/0.5) | no | 0.41 (7/0.75/0.01) |
| Janssonhaugen BH10 | Svalbard | 275 | ice-poor | bedrock | −4.5 (−4.8/−4.1) | cold | 0.22 (10/0.06/0.51) |
| Jungfrau | W Alps | 3590 | ice-poor | bedrock | −4.8 (−5.3/−4.4) | cold | 0.65 (10/0.7/0.00) |
| Juvvasshoe BH1 | Scand. S | 1851 | ice-bearing | blockfield | −1.9 (−2.1/−1.8)* | warm | 0.17 (9/0.25/0.17) |
| Juvvasshoe BH3 | Scand. S | 1546 | ice-poor | debris | −0.4 (−0.7/0.2) | no | 0.89 (7/0.93/0.00) |
| Juvvasshoe BH30 | Scand. S | 1894 | ice-poor | blockfield | −2.5 (−2.9/−2.4)* | cold | 0.56 (9/0.92/0.00) |
| Juvvasshoe BH4 | Scand. S | 1547 | ice-poor | bedrock | −0.3 (−0.7/−0.1) | near-zero | 0.64 (7/0.81/0.01) |
| Lapires BH1 | W Alps | 2500 | ice-bearing | debris | −0.1 (−0.1/−0.1) | near-zero | 0.02 (10/0.25/0.14) |
| Lapires BH11 | W Alps | 2500 | ice-bearing | debris | −0.1 (−0.3/0) | near-zero | 0.17 (7/0.61/0.07) |
| Lapires BH12 | W Alps | 2535 | ice-bearing | debris | −0.1 (−0.3/−0.1) | near-zero | 0.10 (10/0.25/0.17) |
| Les Attelas BH1 | W Alps | 2661 | ice-bearing | debris | −0.7 (−0.8/−0.5) | warm | −0.10 (10/0.11/0.34) |
| Les Attelas BH2 | W Alps | 2689 | ice-bearing | debris | −0.2 (−0.3/−0.1) | near-zero | 0.05 (8/0.06/0.57) |
| Matterhorn Hoernlih. | W Alps | 3314 | ice-poor | bedrock | −1.4 (−1.7/−1.1) | warm | 0.54 (9/0.76/0.00) |
| Matterhorn Ridge | W Alps | 3343 | ice-poor | bedrock | −2.5 (−2.8/−2.3) | cold | 0.56 (5/0.89/0.02) |
| Muot d. Barba P. BH1 | E Alps | 2946 | ice-poor | debris | −1.1 (−1.3/−0.9) | warm | 0.04 (9/0.01/0.84) |
| Muot d. Barba P. BH2 | E Alps | 2942 | ice-poor | debris | −0.8 (−1.1/−0.6) | warm | 0.20 (9/0.10/0.40) |
| Muragl BH1 | E Alps | 2549 | no ice | coarse bl. | 0.9 (0.8/1.2) | no | 0.16 (7/0.11/0.47) |
| Murtèl-Corvatsch 87 | E Alps | 2670 | ice-rich | coarse bl. | −1.2 (−1.6/−0.8) | warm | 0.19 (9/0.04/0.60) |
| Piz Boe | W Alps | 2900 | ice-poor | bedrock | −0.1 (−0.4/0.4) | no | 0.46 (9/0.27/0.15) |
| Ritigraben | W Alps | 2634 | ice-rich | coarse bl. | −0.3 (−0.4/−0.2) | near-zero | −0.09 (11/0.27/0.12) |
| Saudhafell | Iceland | 906 | ice-bearing | debris | −0.3 (−0.6/−0.3)* | near-zero | 0.27 (8/0.74/0.01) |
| Schafberg BH1 | E Alps | 2754 | ice-rich | coarse bl. | −0.4 (−0.7/−0.3) | near-zero | 0.22 (9/0.19/0.24) |
| Schafberg BH2 | E Alps | 2732 | ice-rich | coarse bl. | −0.1 (−0.2/−0.1) | near-zero | 0.05 (8/0.45/0.07) |
| Schilthorn 5000 | W Alps | 2909 | ice-poor | debris | −0.1 (−0.4/−0.1)* | near-zero | 0.25 (9/0.74/0.01) |
| Schilthorn 5198 | W Alps | 2910 | ice-poor | debris | −0.1 (−0.2/0.1) | no | 0.26 (10/0.71/0.00) |
| Schilthorn 5200 | W Alps | 2909 | ice-poor | debris | −0.1 (−0.3/0.2) | no | 0.51 (10/0.89/0.00) |
| Sonnblick | E Alps | 3075 | ice-poor | bedrock | −1.6 (−1.8/−1.5) | warm | 0.24 (8/0.60/0.02) |
| Stelvio | E Alps | 3000 | ice-poor | bedrock | −1.1 (−1.5/−0.7) | warm | 0.67 (8/0.43/0.08) |
| Stockhorn 6000 | W Alps | 3412 | ice-poor | bedrock | −1.7 (−2/−1.4) | warm | 0.52 (10/0.65/0.01) |
| Stockhorn 6100 | W Alps | 3410 | ice-poor | bedrock | −0.7 (−0.8/−0.6) | warm | 0.22 (10/0.63/0.01) |
| Tarfalaryggen | Scand. N | 1550 | ice-poor | bedrock | −2.0 (−2.7/−1.7)* | cold | 0.93 (8/0.85/0.00) |

**Table 1 (continued) | Boreholes in European mountains in permafrost areas and 2013–2022 warming rates at 10 m depth**

| Borehole name | Region | Elev. (m asl.) | Ground ice content | Surface cover | MGT$_{10yr}$ (min/max MGT$_{1yr}$) (°C) | Permafrost conditions | Warming rate (°C dec$^{-1}$) (n/R$^2$/p) |
|---|---|---|---|---|---|---|---|
| Tron | Scand. S | 1628 | ice-poor | blockfield | 0.2 (−0.1/0.6) | no | 0.76 (7/0.97/0.00) |
| Veleta Peak | Sierra N. | 3380 | no ice | bedrock | 2.4 (2.1/3.0) | no | 0.87 (7/0.65/0.03) |
| Vopnafjordhur | Iceland | 892 | no ice | debris | 0.5 (0.2/0.8)* | no | 0.44 (8/0.28/0.18) |
| Zugspitze N | E Alps | 2922 | ice-poor | bedrock | −1.9 (−2.2/−1.6) | warm | 0.61 (10/0.79/0.00) |
| Zugspitze S | E Alps | 2922 | ice-poor | bedrock | 0.7 (0.5/0.8)* | no | 0.36 (10/0.67/0.00) |

If the temperature time series did not extend to 2022, the latest available decadal period (ending no later than 2018) was considered (*). Regions abbreviations are Alps E and W for Eastern and Western Alps, Scand. N and S for Scandinavia North and South, and Sierra N. for Sierra Nevada. Elev. is the elevation, MGT$_{10yr}$ is the mean ground temperature of the latest decade (with minimum and maximum annual ground temperature MGT$_{1yr}$ of the decade in brackets). The number (n) of MGT$_{1yr}$ values available to calculate the warming rate is given together with the coefficient of determination (R$^2$) and the p-value (p). Five of the boreholes shown in Figs. 1 and 2 are not shown here as no recent decadal warming rate could be calculated at 10 m depth. In boreholes of the class no permafrost but with indicated ground ice, the permafrost has disappeared at 10 m depth during the observation period (maximum MGT$_{1yr}$ > 0 °C). Gemsstock N & S and Zugspitze N & S are the two halves of a horizontal borehole through a rock ridge. Details on the boreholes can be found in Supplementary Table 1.

permafrost is cold or absent ($n = 10$, 90% of p-values < 0.05). Warming rates can also be high in non-bedrock slopes where permafrost has disappeared during the 10-year period. In fact, the maximum value of 1.77 °C dec$^{-1}$ was observed for Dovrefjell DB6 in southern Norway. Warming rates lower than 0.5 °C dec$^{-1}$ are typically observed in ice-bearing talus slopes and blockfields or in rock glaciers ($n = 25$, 44% of p-values < 0.05). Cooling rates were obtained for 4 locations in warm or near-zero permafrost in ice-bearing ground. They are, however, lower than 0.1 °C dec$^{-1}$ (p-value < 0.05 only for one time series).

At depths greater than 10 m and over a longer time period, calculated permafrost warming rates are generally lower and statistically more robust: at 20 m depth they range up to 0.71 °C dec$^{-1}$ for 2013–2022 (mean 0.24 °C dec$^{-1}$, $n = 33$, 73% of p-values < 0.05) and for the 20-year period 2003–2022 at 10 m depth up to 0.85 °C dec$^{-1}$ (mean 0.39 °C dec$^{-1}$, $n = 16$, 94% of p-values < 0.05). A non-parametric Wilcoxon signed rank test on paired samples of warming rates at 10 m and 20 m depth shows a significant difference (0.05 level) for both 10-year ($n = 35$, mean difference 0.13 °C dec$^{-1}$) and 20-year periods ($n = 13$, mean difference 0.12 °C dec$^{-1}$). This shows a delayed propagation of the recent acceleration of the warming to greater depths resulting from slow heat conduction. For the time period 2007–2016, Biskaborn et al.[3] found in the global-scale analysis an average mountain permafrost warming rate of 0.19 °C dec$^{-1}$ at the DZAA (mean of 14 m). Based on the time series in our dataset that cover the period 2007–2016, we obtain a mean warming rate of 0.39 °C dec$^{-1}$ at 10 m depth ($n = 26$, 65% of p-values < 0.05) and of 0.24 °C dec$^{-1}$ at 20 m depth ($n = 20$, 85% of p-values < 0.05) for 2007–2016, and slightly higher values of 0.40 and 0.26 °C dec$^{-1}$ for 10 and 20 m depth (64% and 84% of p-values < 0.05) for 2013–2022, respectively.

Warming rates vary between time periods depending on the inter-annual fluctuations of MGT$_{1yr}$. This variability is shown by warming rates over a moving window of 10 and 20 years (Fig. 3, Supplementary Fig. S4). At 5 m depth, the warming rates can fluctuate by more than 1.5 °C dec$^{-1}$ when moving a 10-year averaging window by 5 years. They are thus not suited to describe long-term changes in the permafrost. Decadal warming rates for 10 and 20 m depth exhibit lower variation and higher significance, but can still vary up to nearly 1.0 and 0.5 °C dec$^{-1}$, respectively. Considering 20-year periods, the variability is clearly lower than for decadal warming rates. Figure 4 shows the most recent values (cf. diamonds) in the context of the variability of decadal warming rates at 10 m depth for all boreholes of the dataset. This variability is however influenced by the length of the time series. At the coldest sites and for some sites in Svalbard, for example, most recent decadal warming rates are lower than those obtained for earlier 10-year periods.

## Permafrost warming patterns

For the latest available decade of data, mean permafrost warming rates at 10 m depth are 0.36 °C dec$^{-1}$ in the European Alps ($n = 34$), 0.40 °C dec$^{-1}$ for Svalbard ($n = 2$), 0.63 °C dec$^{-1}$ for Scandinavia ($n = 11$), and 0.2 °C dec$^{-1}$ for Iceland ($n = 3$). Differences related to the thermal conditions (cf. permafrost classes), and ground ice content surpass these regional differences (Fig. 5). We observe a general decrease in warming rates with ground temperatures increasing towards 0 °C (Fig. 6). Cold permafrost warms at a mean rate of 0.67 °C dec$^{-1}$ ($n = 9$), compared to 0.35 °C dec$^{-1}$ for warm permafrost ($n = 13$) and 0.13 °C dec$^{-1}$ for near-zero permafrost ($n = 16$). Ground temperatures at sites where permafrost is absent or has disappeared during the considered time period are increasing by 0.60 °C dec$^{-1}$ on average ($n = 21$). Warming rates in cold permafrost, at ice-poor bedrock sites and at non-permafrost sites are in a similar range and correspond to warming rates of surface air temperature (SAT) over the past 30 years (Fig. 5, cf. 'Methods' for details).

The classification by ground ice content (cf. 'Methods') confirms this pattern: For ice-rich rock glaciers, warming rates at 10 m depth are very low (0.05 °C dec$^{-1}$, $n = 6$, 17% of p-values < 0.05) and ground temperatures can remain just below 0 °C for many years until the ground ice has melted at the respective depth (e.g., Ritigraben, Schafberg, Bellecombes). For ice-bearing sites like talus slopes or blockfields, the mean warming rate is 0.34 °C dec$^{-1}$ ($n = 14$, 43% of p-values < 0.05). Once the ground ice has melted, warming rates substantially increase as the buffering effect of latent heat is no longer effective (cf. thawed sites in Fig. 4 and sites which originally contained ice with MGT$_{10/20yr}$ > 0 °C in Fig. 6). Sites with little or no ground ice typically warm at rates of >0.5 °C dec$^{-1}$ (the mean warming rate for ice-poor sites is 0.51 °C dec$^{-1}$ with $n = 30$ and 77% of p-values < 0.05 and the mean warming rate for the sites without ice is 0.55 with $n = 9$ and 44% of p-values < 0.05). The classification following surface cover characteristics also yields a similar pattern. Surface cover characteristics are typically related to the type of landform and, hence, to the ground ice content.

The patterns described above refer to the latest decade and a depth of 10 m but prove consistent across all considered depths, time periods and time spans (cf. Supplementary Figs. S5–S9).

## Seasonal distribution of permafrost warming

Examination of monthly ground temperatures at 10 m depth reveals distinct differences in warming rates over the course of the year. They are characterized by prominent months for both the highest and lowest values and show a typical spread of 0.2–0.4 °C dec$^{-1}$ (Fig. 7). The patterns observed for annual values based on the classification by permafrost conditions, surface cover and ground ice content also match with those derived for monthly warming rates. On average, the maximum monthly warming rate in cold permafrost was 0.74 °C dec$^{-1}$. The highest monthly warming rates are observed at 10 m depth at Aiguille du Midi NE in the Western Alps (monthly warming rate of 1.1 °C dec$^{-1}$ in November, with a phase lag of five months with respect to the ground surface, cf. Supplementary Table S1) and at Janssonhaugen

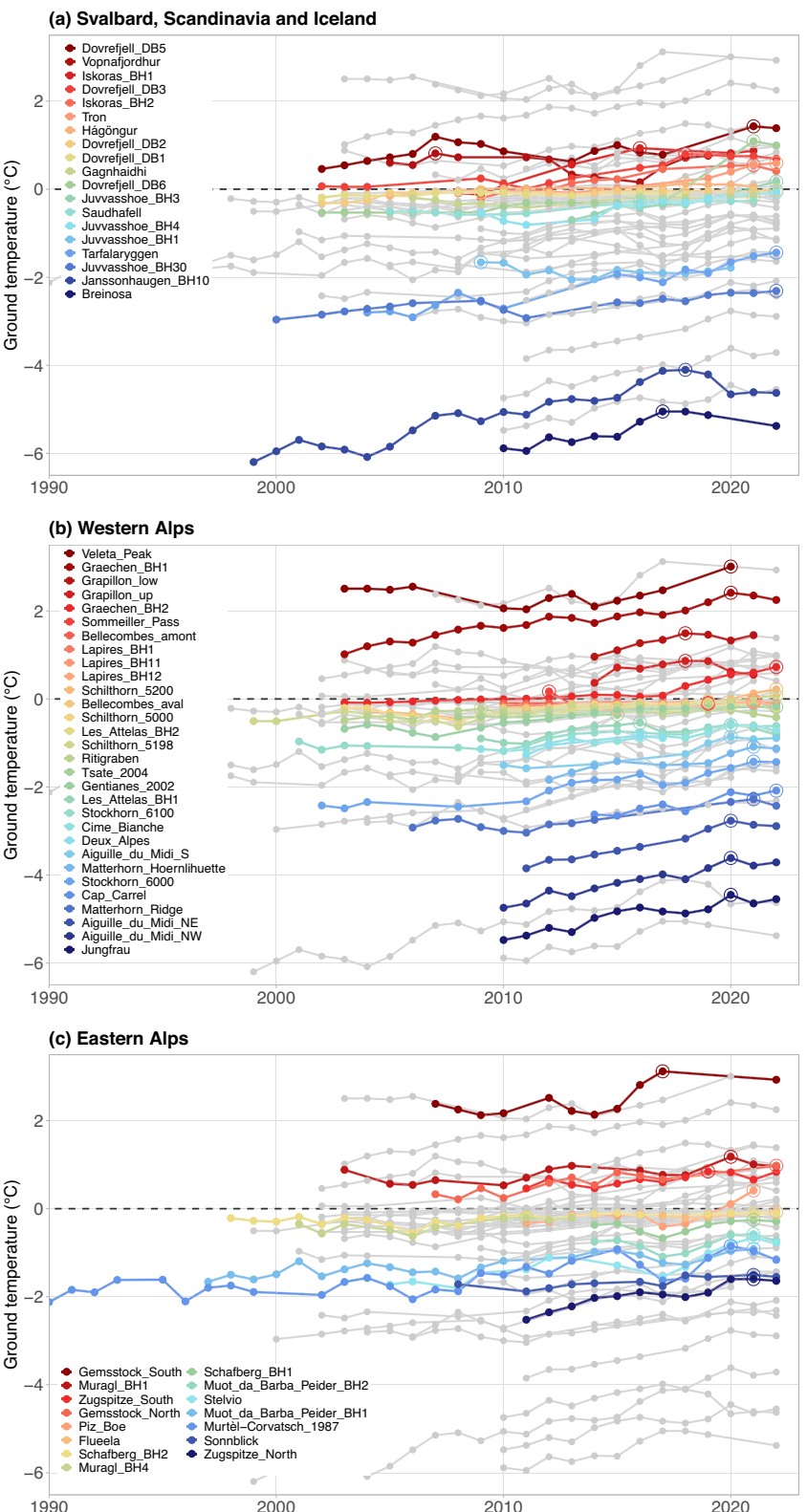

**Fig. 2 | Sixty-four records of annual ground temperatures obtained in or next to permafrost areas in European mountains at approximately 10 m depth that cover at least one decade. a** Svalbard, Scandinavia (sites in mainland Norway and Sweden) and Iceland, (**b**) Western Alps (sites in France, Western Switzerland, North-western Italy, and including the site in the Spanish Sierra Nevada), and (**c**) Eastern Alps (sites in Central and Eastern Switzerland and North-eastern Italy, Germany and Austria). The circles indicate the maximum value of each time series. In each panel, the time series of the other regions are shown in grey for comparison. The records obtained at 20 m depth are shown in Supplementary Fig. 3.

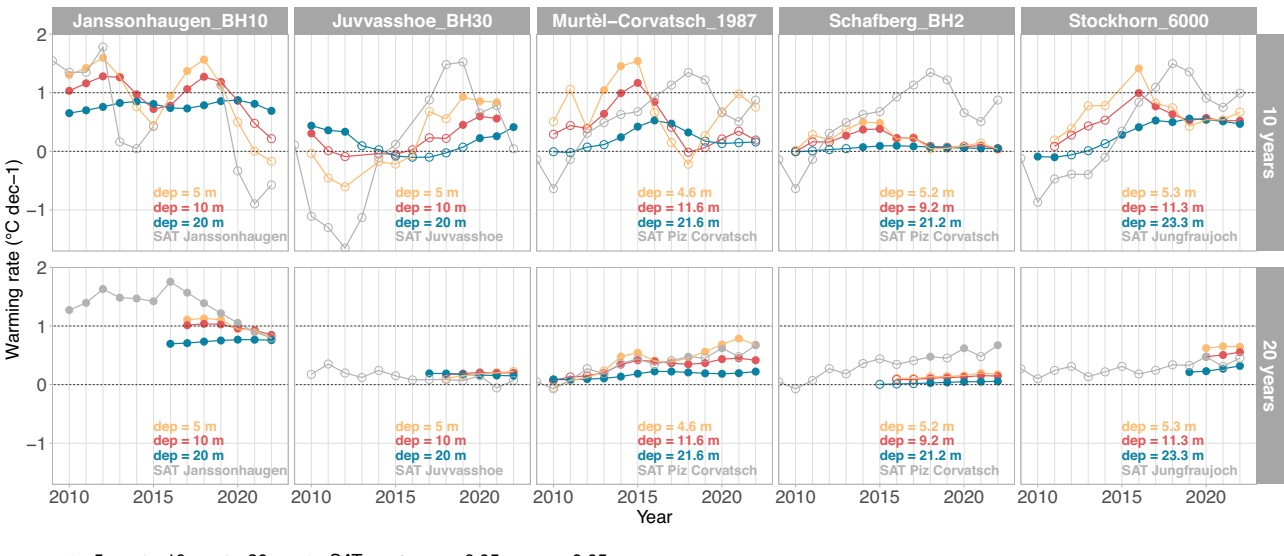

**Fig. 3 | Examples of running warming rates for time series of annual mean permafrost temperatures with an averaging window of one (top) and two (bottom) decades.** Running warming rates for surface air temperatures (SAT) obtained nearby are displayed for comparison (data source: Federal Office for Meteorology and Climatology MeteoSwiss, Norwegian Meteorological Institute, The University Centre in Svalbard). Depths of 5, 10 and 20 m as well as SAT are distinguished by colours, significance of the warming rates at 0.05 level is distinguished by open or solid circle symbols. Values are plotted at the end of the period, e.g. the warming rates for 2013–2022 are shown at $x = 2022$. Actual depths of the measurements and the name of the weather station are given at the bottom of each plot. Running warming rates for all permafrost temperature time series included in the study are shown in Supplementary Fig. 4.

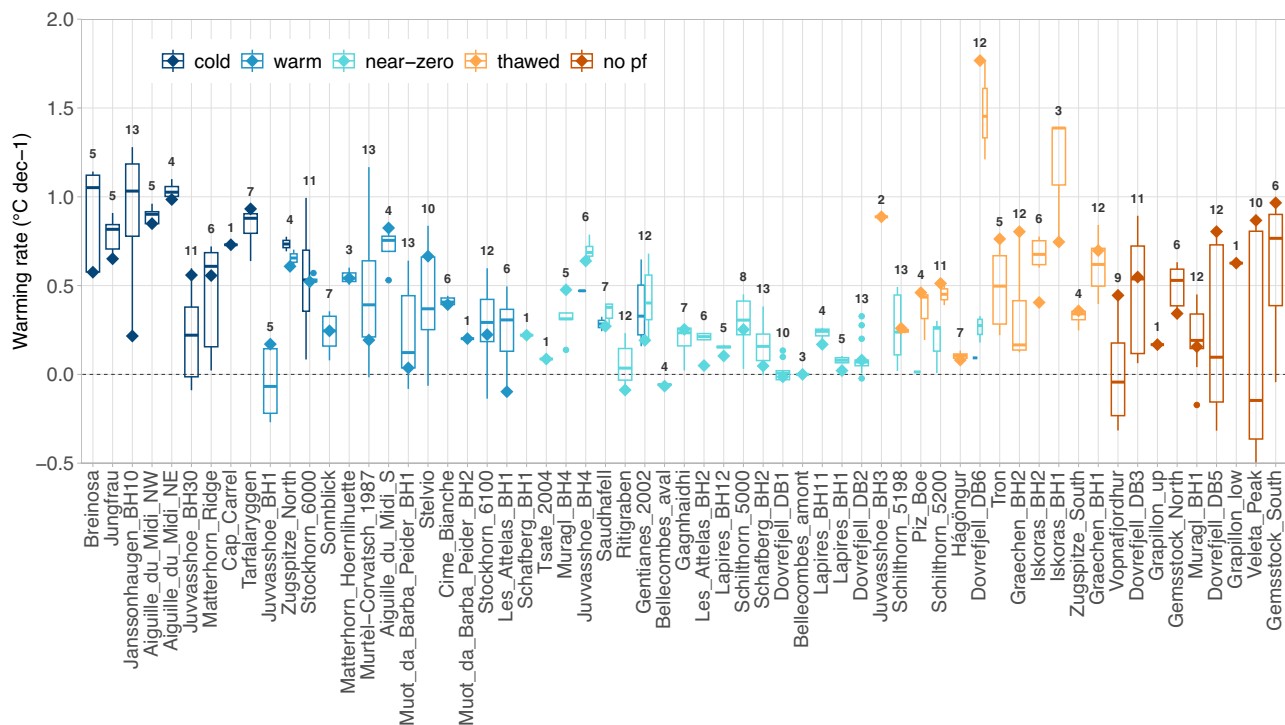

**Fig. 4 | Variability of decadal warming rates for all permafrost temperature time series.** The boxplots show warming rates at 10 m depth calculated from ground temperature measurements in permafrost areas in European mountains for all available 10-year periods starting from 2000. Diamond symbols show the warming rate for the latest available decade (2013–2022 or max. 4 years earlier, cf. Table 1). The number of calculated warming rates per borehole time series is indicated on top of the boxes. Boreholes are sorted by their mean ground temperature during the last decade ($MGT_{10yr}$), with the coldest to the left. The colours indicate the permafrost conditions (no pf indicates the nine non-permafrost boreholes and thawed are boreholes that were in permafrost at the beginning but not at the end of the latest decade). If permafrost conditions changed during the observation period, two narrow boxplots in different colours are shown for the same borehole.

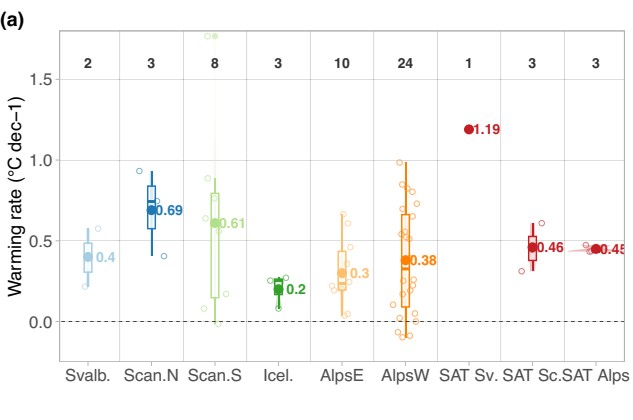
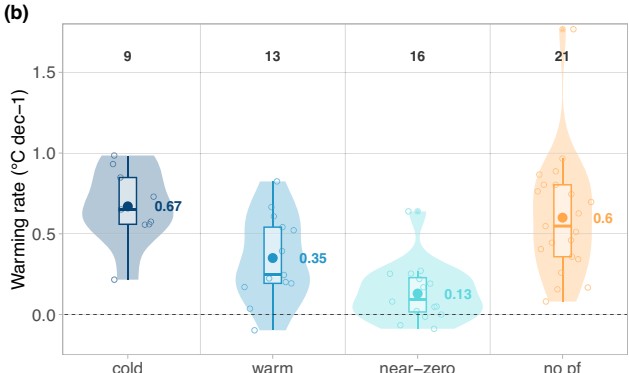
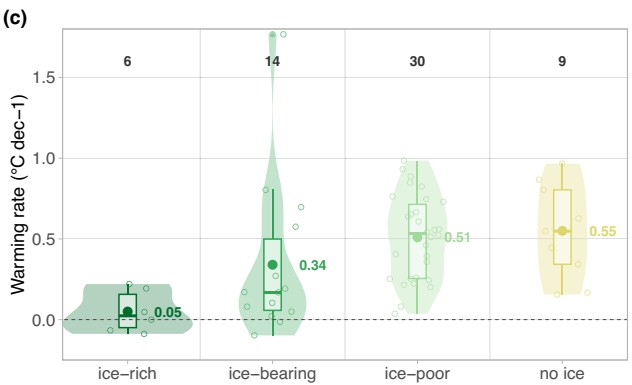
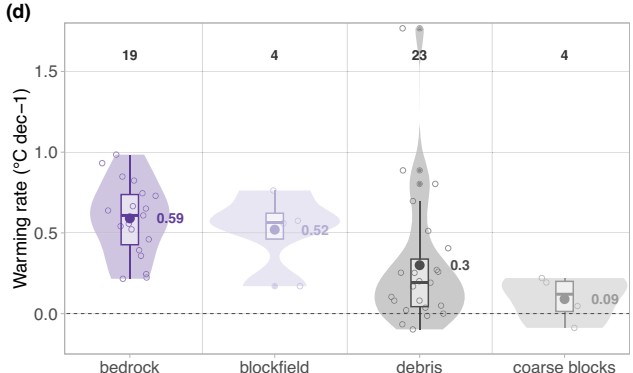

**Fig. 5 | Permafrost warming rates at 10 m depth for the latest available decade, classified by region, permafrost conditions, ground ice content and surface cover. a** Recent decadal warming rates (2013–2022 or max. 4 years earlier) at 10 m depth for six European mountain regions (Svalbard, Scandinavia North, Scandinavia South, Iceland, Eastern Alps and Western Alps) and compared to surface air temperature SAT warming rates for the 30-year period 1993–2022 in Svalbard (SAT Sv.), Scandinavia (SAT Sc.) and the European Alps (SAT Alps). **b–d** Warming rates classified by permafrost conditions, ground ice content, and

surface cover. In all panels, the distribution for each class is shown by violins and overlaying boxplots, individual data points are shown by circles, and mean values are shown by filled points and in °C dec⁻¹ next to them. The number of time series for each class is given at the top. The nine boreholes not in permafrost are labelled *no ice* in (**c**), are not included in (**a**) and (**d**), and are part of the class *no pf* in (**b**). Source of SAT data: Federal Office for Meteorology and Climatology MeteoSwiss, Norwegian Meteorological Institute, The University Centre in Svalbard.

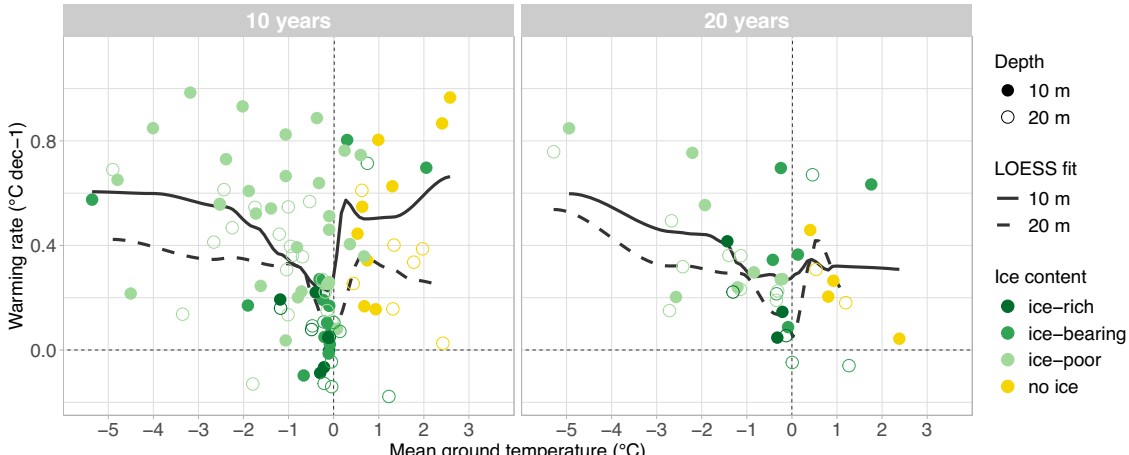

**Fig. 6 | Relation of warming rates to the mean ground temperature of the period.** Ground temperature warming rates at 10 and 20 m depth for the periods 2013–2022 (10 years, left) and 2003–2022 (20 years, right) are plotted against the mean ground temperature during the considered period (10-year mean $MGT_{10yr}$, and 20-year mean $MGT_{20y}$). The black lines show a non-parametric fit (zero degree LOESS fit with span = 0.4) to illustrate the relation between the two quantities at

10 m (solid line) and 20 m depth (dashed line). The highest warming rate over 10 years at 10 m depth (1.77 °C dec⁻¹) lies outside the plot area but is considered for the LOESS fit. Yellow points with no ground ice (labelled no ice) are time series not in permafrost conditions for the entire period and at the respective depth. Green points with positive $MGT_{10yr/20yr}$ are sites where permafrost disappeared at the respective depth during the 10-year (left) or 20-year (right) period.

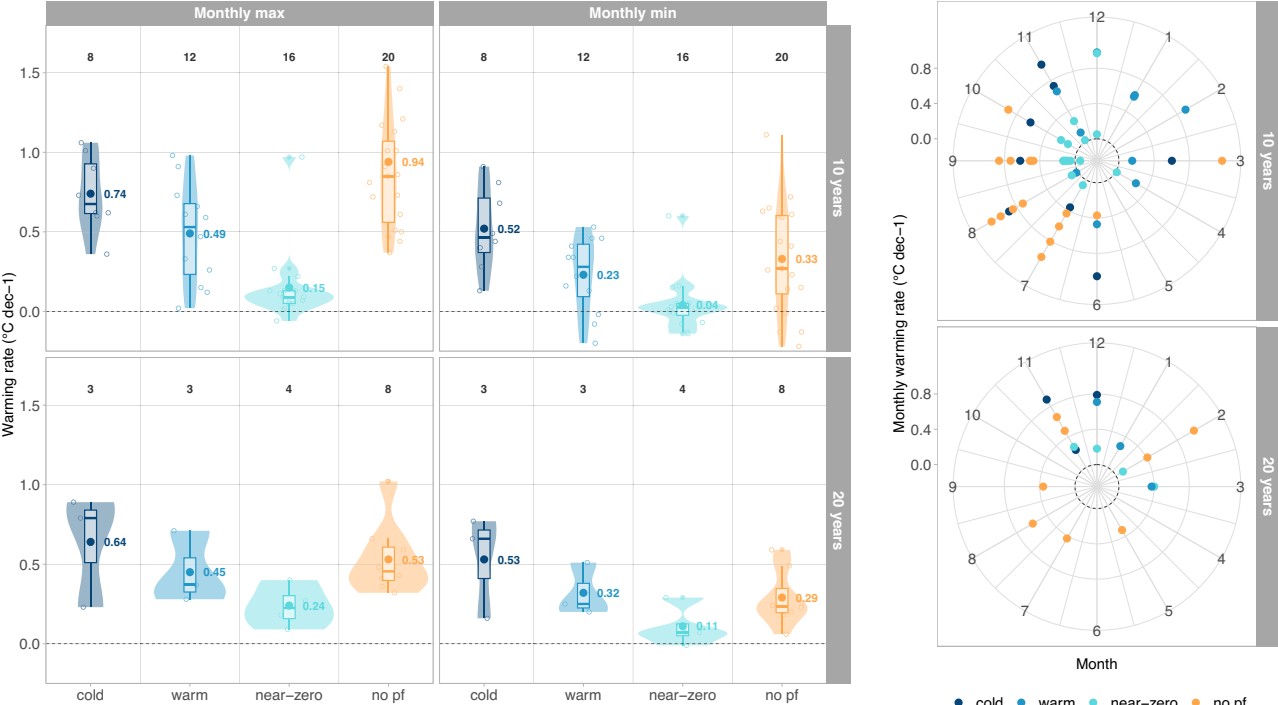

**Fig. 7 | Seasonal distribution of permafrost warming rates.** Left panels: Monthly maximum (left) and minimum (right) warming rates at 10 m depth for the last available 10-year (top) and 20-year (bottom) periods. Permafrost classification (colours), points and boxplots are given as in Fig. 5. Right panels: Maximum monthly warming rate observed at 10 m depth for each time series over the last available 10-year (top) and 20-year (bottom) periods plotted for the month in which they originate near the ground surface, considering the site-specific time delay necessary to reach 10 m depth (cf. Supplementary Table 1). The highest value in the top plot (10 years) amounts to 2.6 °C dec⁻¹ in July and is outside the plot area.

BH10 in Svalbard (monthly warming rate of 0.9 °C dec⁻¹ in June, with a phase lag of six months with respect to the ground surface).

Notable differences emerge between the most recent decade 2013–2022 and the 20-year period 2003–2022 (Fig. 7, left). For the latest decade, the highest monthly warming rates at 10 m depth originate from increased ground surface temperatures in summer and autumn, while for the 20-year period, the highest monthly warming rates are primarily influenced by winter warming (Fig. 7, right). For the 10-year period, summer and autumn contribute equally to warming for cold and warm permafrost, while for near-zero permafrost, autumn has a more significant impact. In areas without permafrost, summer is the dominant warming season. Over the 20-year period, winter emerges as the dominant warming season for cold-, warm and near-zero permafrost. Regionally, early winter appears as the peak warming time in Scandinavia and Svalbard, while late winter is most dominant in the Alps. Finally, non-permafrost areas experience the highest warming during summer and autumn.

## Discussion

Permafrost in European mountains has generally warmed since the start of ground temperature monitoring. This is observed across all regions, depths and time periods considered in our analyses. Temperature changes in the permafrost are primarily driven by changes in SAT[57]. Observed warming patterns depend mainly on the depth of observation and on the amount of latent heat exchanged in the ground. The latter depends on the ground temperature and the ground ice/water content.

Warming rates for the latest available decade are generally higher than reported earlier for mountain permafrost regions[3,4,6,8,50]. Most likely, this results from the considered sample, since these previous studies mainly included earlier data from warm and near-zero permafrost in sedimentary landforms. Time series from cold high-

elevation bedrock sites, where the warming is strongest, have only just reached a decade and thus the minimum length to evaluate their changes. A fast temperature increase in near-vertical bedrock slopes, ridges and peaks can be accelerated by steep Alpine topography, as the rock can be warmed from several mountain sides[58].

Most recent ground temperature change rates at a depth of 10 m in cold permafrost, in ice-poor bedrock slopes and in adjacent terrain with no permafrost are in a range comparable to SAT change rates derived for the European Alps and Scandinavia[59] (Fig. 5). Since the 1980s when the first mountain permafrost data were measured, there was a marked atmospheric warming in the Alps, which appears to have become more pronounced in recent years[60]. The highest rates of SAT increase are observed in summer, particularly at high elevations (>2000 m asl.)[60–62]. In the Scandinavian mountains, SAT warming over recent decades has not been systematically investigated to date, but we derived values comparable to the European Alps (Fig. 5). In western Svalbard, 1981–2020 SAT warming rates between 1.0 °C and 1.2 °C dec⁻¹ were observed near the permafrost monitoring sites[56].

Warming rates generally decrease when ground temperatures approach the melting point, and are then considerably lower than those of SAT. Just below 0 °C, ground temperatures in ice-rich terrain can remain fairly stable for years, despite changing SAT[4,8,11]. However, low warming rates in warm and near-zero permafrost do not imply the absence of changes in the permafrost. On the contrary: in ice-bearing and ice-rich ground, large changes in frozen and unfrozen water content take place in this temperature range, where ice and water can co-exist[63]. This can critically impact the geotechnical and hydrological ground properties. These changes, however, are not reflected in ground temperatures as additional heat is absorbed for phase change until the ice has melted, but can be captured by complementary methods like repeated geophysical surveys. Such techniques revealed substantial changes in unfrozen water content at many permafrost

sites in European mountains, pointing to a substantial and irreversible loss of ground ice[54,64–66]. After the permafrost thaws, ground temperature response can be abrupt, leading to very high warming rates. Indeed, the highest warming rate in this study is related to a sudden increase in ground temperature in the borehole Dovrefjell DB6 in 2017, which occurred after permafrost disappeared completely. Observed cooling rates are below 0.1 °C dec⁻¹ for the latest decade and occur at sites with permafrost temperatures close to 0 °C, where inter-annual variations are in the range of measurement accuracy. We therefore do not consider the cooling to be of actual significance.

Depending on the measurement depth, the permafrost temperature series exhibit considerable intra- and interannual variations. Permafrost temperature changes at around 10 m depth reflect the systematic seasonal variations at the ground surface but with a smaller amplitude and a phase lag as higher-frequency near-surface signals are filtered out. This poses challenges in deriving statistically significant warming rates for shorter time periods such as a decade, particularly when using annually aggregated data. The significance of the warming rates (percentage of $p$-values < 0.05) increases with the length of the period, the depth considered and with the magnitude of the warming rate (cf. 'Methods'). $P$-values > 0.05 for the latest decadal warming rates at 10 m depth are mostly obtained for sites with low warming rates or where short-term variability was higher, e.g. due to a cooling phase following snow-poor winters (e.g. Janssonhaugen BH10 or Murtèl-Corvatsch 1987). For these sites, however, statistically significant warming rates were derived for other periods for all locations except two (cf. Supplementary Fig. S4). Excluding warming rates with $p$-values > 0.05 would therefore not only reduce the number of observations but result in an overestimation of mean warming rates, particularly for lower values. We also tested the resulting warming pattern based on warming rates with $p$-values < 0.05 only and found that they are not controlled by non-significant values.

The results from the seasonal analyses correspond to the previously reported influence of SAT, radiation and snow cover changes on the permafrost thermal regime as well as to modelling results of permafrost sensitivity to seasonal SAT and precipitation anomalies[67]. Particularly for the last decade, the higher summer and autumn SAT in Europe[59,68] caused by increased surface solar radiation[69,70], seem to have contributed importantly to permafrost warming. This is particularly true for summer 2022, which was the warmest summer on record in Europe and for which highest annual surface solar radiation levels were observed[69]. Distribution and changes of mountain permafrost further depend on net radiation[71], which leads to a different thermal response at different slope expositions in high-elevation bedrock[11]. However, local to regional variations have to be considered when transferring the overall response to individual sites. The aforementioned effects influence the sites to varying degrees[72] and contribute to explaining the differences in warming rates observed at our monitoring sites.

Further, we consider the timing and duration of the winter snow cover, which can temporarily reduce or enhance a general warming pattern. An early snow cover typically conserves the summer heat in the ground, while a long-lasting snow cover insulates the ground from increasing SAT in spring or early summer. Changes in SAT during winter have most effect on ground temperatures during periods with little or no snow. A temporary cooling of the permafrost to more than 10 m despite higher SAT was for example observed following the snow poor winters 2016 and 2017 in the European Alps[8,73] (cf. Figure 3). Snow cover duration in the Alps decreased in recent decades even at the very high elevations, particularly due to an earlier snowmelt in spring[74,75]. A longer snow-free season in spring and early summer has the potential to amplify non-linear processes, such as albedo feedbacks, and to accelerate permafrost warming by corresponding changes in ground heat fluxes[72]. For the analysed 20-year period, near-surface warming was most dominant in winter (Fig. 7), in agreement with long-term

changes in SAT and snow cover. In Scandinavia, an increase in winter precipitation since 1961 was observed, resulting in greater snow accumulation in the highest mountains[76] and contributing to permafrost degradation[54]. In Svalbard, the highest SAT warming rates were observed during autumn and winter[56]. Here, monitoring sites are snow poor, making the late autumn and winter warming's impact on the permafrost strongest, as indicated in our seasonal analysis.

Permafrost observation sites are geographically unevenly distributed with a bias toward more accessible locations or pre-existing research sites[12]. While mountain permafrost research was initiated in the 1970s with studies on prominent rock glacier landforms[43], permafrost in near-vertical rock has only been in the focus since the 2003 summer heat wave, during which rockfall activity was high[77]. Several additional boreholes have recently been drilled into high-elevation bedrock, but the related time series are yet too short (<1 decade) to evaluate long-term changes. The information thus remains very scarce for the highest elevations (e.g. 4000 m asl. and higher in the Alps), where permafrost temperatures are likely a few degrees lower than those available to date. In Europe, the eastern end of the Alpine Arc, the Pyrenees, the Carpathians, and the remote areas in Scandinavia and Svalbard are underrepresented. At the larger scale, mountain permafrost temperatures are monitored in the Qinghai-Tibet Plateau e.g., ref. 35. Otherwise, however, mountain permafrost temperature data is rare and there are only few observations in the Rocky Mountains, the Central Asian mountain ranges, the Himalaya and the Andes.

Maintaining temperature sensors and data loggers in harsh environments over decades to ensure uninterrupted, robust and comparable records of permafrost temperature poses significant challenges. Standards for site selection and measurement protocols for permafrost temperatures have only recently been elaborated[38,78]. The World Meteorological Organization (WMO) approved the first global standardization of permafrost monitoring in their WMO Guide for Instruments and Methods of Observation in 2024[79]. Maintenance work will likely increase in the future with changing permafrost conditions at the sites, and some sites are becoming difficult to access due to field safety concerns. The continuation of the measurements often relies on the principal investigators and their financial support. A large part of the time series included in this study are not secured for the long term and only a part of the time series in our dataset is centrally managed in an (inter)national database. Our study underlines the importance of maintaining uninterrupted long-term records of permafrost temperatures for the assessment of climate change impacts. The need for national and international efforts for permafrost monitoring, data curation and sharing, and for updated reports has recently been highlighted by the WMO[80] and in international political frameworks[81]. Such monitoring is vital for assessing the impact of atmospheric warming on permafrost, as well as for improved model validation, stakeholders, and addressing challenges posed by permafrost degradation in mountain regions. Permafrost warming at depth is a relatively slow yet inexorable process. The higher warming observed in the upper 10 metres of the ground compared to 20 m depth points to an increasing thermal disequilibrium between the uppermost metres and permafrost at depth. For the coming decades, the propagation of these changes to greater depths is largely predetermined[82].

## Methods

### Measurement of permafrost temperatures

Ground temperature time series in mountain permafrost are obtained in boreholes of around 10–100 m depth, which are drilled vertically in most cases, but also horizontally through crests or perpendicular to the surface in near-vertical bedrock slopes. Temperatures are continuously and automatically logged at different depths with permanently installed multi sensor strings and at intervals of 1–24 h. Thermistors are the most widely used sensors for permafrost

temperature measurements. Temperature sensor strings are typically calibrated in an ice-water bath at 0 °C with the logging system prior to installation in the borehole. The drilling and instrumentation of boreholes in harsh, cold environments with difficult access, and their long-term operation over decades are technically and logistically challenging. Guidelines on the measurement procedure were recently published[38,78,79].

Typical measurement accuracies reported for permafrost temperature measurements vary depending on the sensor types and logging system between ±0.01 and ±0.25 °C[3,38,78]. Gaps in the time series can occur due to power failure or damage from moisture intrusion, natural hazards (storms, lightning, avalanches, rock falls), or disruption by animals. Potential data inconsistencies may result from changes in the measurement setup (such as replacement of sensor strings and loggers, or change of maintenance procedures), sensor drift over time, moisture or water intrusion into the borehole or instruments (particularly for sites with temperatures close to 0 °C with higher unfrozen water contents), or damage due to ground deformation. Re-calibration of the system to assess sensor drift is often not possible as sensor strings can be stuck in the borehole, particularly in ice-rich deforming ground such as rock glaciers[83]. Data quality control primarily relies on value-range and consistency checks, as well as on plausibility assessment based on data from nearby sensors or stations. A single drifting sensor is detectable through its anomalous temporal trend compared to neighbouring sensors.

## Data collection and processing
Ground temperature time series measured in boreholes in European mountain permafrost regions were collected from national networks such as the Swiss Permafrost Monitoring Network (PERMOS, www. permos.ch), the Norwegian Permafrost Monitoring Network (cryo.met. no and sios-svalbard.org), and the Réseau français d'observation du permafrost (PermaFrance, permafrance.osug.fr). Additional collected data from regions without a national repository were maintained on the long-term by research institutions (such as the Universities of Barcelona, Insubria, Oslo and Stockholm) and regional environment agencies (such as GeoSphere Austria, the Regional Environmental Protection Agencies in Northern Italy or the Bavarian Environment Agency). Many of these networks and institutions contribute to the Global Terrestrial Network for Permafrost (GTN-P, gtnp.arcticportal. org). Detailed data sources for each time series are provided in Supplementary Table S1.

The criteria to include a ground temperature time series in this study were: (i) the borehole is in permafrost region in mountain terrain ($MGT_{1yr} \leq 0$ °C for two consecutive years), (ii) reaches a depth of at least 10 m, and (iii) the time series covers at least one decade until 2022. For comparison to changes in non-permafrost terrain, we additionally considered data from boreholes located in close vicinity of permafrost.

We performed basic quality checks on daily ground temperature time series and consistently aggregated them to monthly ($MGT_m$) and annual mean values ($MGT_{1yr}$). Due to the decreasing temporal variability of ground temperatures with depth, we define depth-dependent criteria for data completeness: depth ranges with daily fluctuations (above 2 m depth, 12 $MGT_m$ values required to calculate $MGT_{1yr}$), with seasonal fluctuations (2–17 m depth, 10 $MGT_m$ required for $MGT_{1yr}$) or with only intra-annual fluctuations (below the DZAA at ca. 15 m, 4 $MGT_m$ required).

For the analyses of warming patterns, borehole sites were classified based on region, surface cover, permafrost conditions and ground ice content (cf. Table 1, Supplementary Table S1). For the latter, usually only little and qualitative information is available from drilling logs or semi-direct geophysical soundings. This means that only a basic classification related to the amount of ground ice is possible. However, this

can be used for a general assessment of the influence of latent heat exchange during warming. We therefore considered whether ground ice is only present in rock fissures and pores as is the case in bedrock, if there is considerable ground ice as is possible in unconsolidated sediments, or if excess ice is present, as in rock glaciers. We distinguished the four classes *no ice* (referring here to boreholes that are not in permafrost), *ice-poor* (ground ice only in pores or clefts; ground ice content up to around 10 Vol.%), *ice-bearing* (considerable and varying ground ice in unconsolidated sediments, ground ice content about 10–50 Vol.%), and *ice-rich* (ice supersaturation in rock glaciers; ground ice content up to 100 Vol.%).

While surface cover and ground ice content were considered a constant characteristic for a borehole site, permafrost conditions were assigned for each period and depth. In addition to the classical distinction between warm and cold permafrost[4], we considered that the portion of unfrozen water, and hence the exchanged amount of latent heat, increases exponentially when the ground temperature increases towards 0 °C e.g., refs. 84,85. Most of the phase change takes place at temperatures very close to 0 °C. We distinguished the following four classes: *cold permafrost* with a mean ground temperature for the 10-year or 20-year period ($MGT_{10y}$ or $MGT_{20y}$) below −2 °C (negligible latent heat effects expected), *warm permafrost* with $MGT_{10y/20yr} \geq -2$ °C and < −0.5 °C (small latent heat effects, depending on ground freezing characteristics), and *near-zero permafrost* with $MGT_{10y/20yr} > -0.5$ °C and maximum $MGT_{1yr} \leq 0$ °C (considerable latent heat effects during phase change expected for locations containing substantial ground ice). We distinguished *no permafrost* from near-zero permafrost when the maximum $MGT_{1yr}$ is >0 °C. That is, time series for which permafrost was present at the beginning but not at the end of the considered period, were classified as no permafrost (and as *thawed* permafrost in Fig. 4).

## Calculation of permafrost warming rates
With increasing depth, the evolution of the permafrost thermal regime integrates temperature changes registered over a longer time period, over a larger surface area and with increasing delay. In previous work, permafrost warming rates were often derived at the Depth of Zero Annual Amplitude DZAA e.g., refs. 3,4. This depth is particularly important when relying on irregular and infrequent manual temperature readings[6,78,86]. However, modern logging systems record quasi-continuous temperature time series allowing for seasonal fluctuations to be factored out. In addition, the DZAA varies between sites and over time with changing surface temperatures and some boreholes, e.g. in near-vertical bedrock, do not reach this depth. We analysed warming rates at three key depths defined for climate related long-term permafrost observation: 5, 10, and 20 m[78]. Ground temperatures closely follow surface temperature variations at 5 m depth and are characterized by seasonal variations at 10 m depth. The latter are no longer observed at 20 m depth, which is generally below the DZAA. Due to different sensor depths in the individual boreholes, time series were assigned to three depth classes using the sensors closest to the key depths. This assignment was manually adapted for a few boreholes because of longer gaps at the closest sensor or specific site characteristics (e.g., to correct for differences between the borehole depth and the actual distance to the surface in steep terrain). Horizontal boreholes that completely pierce a ridge (i.e., the boreholes on Zugspitze and Gemsstock) were considered as two boreholes, one half on each side.

We calculated warming rates for the three depth classes for each time series of $MGT_{1y}$ using ordinary least squares (OLS) linear regression in the R-environment. We did this for the latest available 10-year and 20-year period (2013–2022 and 2003–2022). To assess how much the obtained warming rates depend on the time period and relate them to values from previous time periods, we further calculated a running warming rate over a 10- and 20-year moving window starting from the

first available year. Thus, we obtained warming rates for all available 10-year and 20-year time periods and for 1–3 depth classes for each borehole. For a non-parameteric data distribution, the Theil-Sen Slope (TSS) is often preferred over the OLS method as it is less sensitive to outliers. The comparison of the results from applying both methods, however, showed marginal differences (mean difference for all warming rates <0.02 °C dec$^{-1}$, median difference <0.001 °C dec$^{-1}$). Exceptions are ice-bearing sites with a sudden temperature increase above 0 °C after the ground ice has melted and the latent heat effect ceases to play a role. In such a case, the OLS method is preferred since such a temperature increase is not an outlier but rather a relevant signal. This approach has also been widely applied to determine permafrost temperature warming rates[3,4,7,8].

Following Biskaborn et al.[3], we applied the following data completeness criteria to calculate a warming rate for a given 10-year or 20-year period: one MGT$_{1y}$ value at the start of the period ( ± 1 year), one at the end of the period ( ± 1 year) and a minimum of 5 (10 years) or 10 values (20 years). It is important to note that while the number of MGT$_{1y}$ values used for a linear regression is relatively small ($n$ of 5–10 for 1 decade, 10–20 for 2 decades), these are not independent individual observations. We argue that the aggregated annual data allow for the application of a linear regression to time series covering one decade. Indeed, each value aggregates daily observations over one year and short-term variations are filtered with increasing depth. Further, due to slow thermal diffusion, ground temperatures at depth are the result of a signal emerging over a longer period of time. While the temporal aggregation has little effect on the obtained warming rates, it can reduce the statistical significance of the results, which should be considered when interpreting p-values. To test this, we additionally derived decadal warming rates based on monthly and daily time series for several locations with $p$-value > 0.05, which yielded significant p-values for most tested time series using monthly data, and for all time series using daily data.

To better understand the seasonal distribution of warming, we examined linear change rates in monthly temperatures for the latest 10-year and 20-year periods (2013–2022 and 2003–2022) near the ground surface (at ca. 0.2 m depth), at 5 m and at 10 m depth. Because some locations lacked data at 5 m depth, 10 m depth was selected for subsequent analyses and the calculation of intra-annual amplitudes. The maximum and minimum monthly values were calculated, along with the corresponding months in which they occurred. To identify the month with the highest change rate near the ground surface, we analyzed the month with the maximum warming rate at 10 m depth and corrected for the corresponding phase lag in relation to the surface (cf. Supplementary Table S1). To do so, the signal speed of the annual temperature wave was determined by calculating the time difference between the average monthly maximum temperature at approximately 0.2 m and 10 m depth.

Time series of annual mean surface air temperature (SAT$_{1yr}$) from weather stations located at or close to several permafrost boreholes were used to derive SAT change rates. To avoid effects of local topography or valleys (such as temperature inversions), we used data from three high-elevation weather stations of the Federal Office for Meteorology and Climatology MeteoSwiss (Piz Corvatsch at 3294 m asl., Jungfraujoch at 3571 m asl. and Weissfluhjoch at 2691 m asl.). In addition, we used three mountain weather stations of the Norwegian Meteorological Institute (Juvvasshøe at 1894 m asl., Fokstugu at 973 m asl. and Iskoras at 591 m asl.) and one station on Svalbard from The University Centre in Svalbard (Janssonhaugen at 275 m asl.). The sites are considered representative for three larger regions: the European Alps, Scandinavian mountains and Svalbard. The high-resolution Norwegian reanalysis named NORA3[87] was employed to extend the Scandinavian and Svalbard SAT datasets (except Fokstugu) to cover 30 years. NORA3 was selected as it is currently the only high-resolution hindcast available to cover both the entire Scandinavian Peninsula and

Svalbard up to 2023. Additionally, it was shown to outperform both the reanalysis ERA5 and the earlier hydrostatic 10-km Norwegian Hindcast Archive (NORA10), especially in complex terrain[87]. For direct comparison with the original instrumental observations and to derive a value at the station locations, the NORA3 data series were adjusted by regression analysis (R$^2$ 0.95–0.97). We applied linear regression using the OLS method as for permafrost time series to derive SAT change rates, but for a period of three decades (1993–2022) due to the higher interannual variability of SAT compared to the naturally filtered ground temperatures. The application of the TSS method led to negligible differences for SAT change rates compared to the OLS method.

## Data availability
The data set of permafrost temperatures in European mountain regions compiled for this study is openly available from the Zenodo repository at https://doi.org/10.5281/zenodo.13628540. Annual permafrost temperature time series for many of the sites are also available in the GTN-P database (http://gtnpdatabase.org). Permafrost data from Switzerland are openly available via the Swiss Permafrost Monitoring Network (https://www.permos.ch/data-portal, https://doi.org/10.13093/permos-2023-1). The Frost API (https://frost.met.no) provides open access to operational permafrost monitoring data in Norway and Svalbard and permafrost monitoring products are accessible at the cryospheric information web portal of the Norwegian Meteorological Institute (https://cryo.met.no). Annual surface air temperatures from the Swiss Alps are available from the Swiss Federal Office for Meteorology and Climatology MeteoSwiss (https://www.meteoswiss.admin.ch/services-and-publications/applications/measurement-values-and-measuring-networks.html), and from Norway and Svalbard they are available from the Frost API (https://frost.met.no).

## Code availability
All analyses were performed in R, version 4.3.2. The R-script to calculate warming rates for 10- and 20-year periods based on the compiled data set is available on the Zenodo Repository: https://doi.org/10.5281/zenodo.13642979.

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

## Acknowledgements

Long measurement series as used in this study were collected over decades by a large number of people and institutions, as part of regional or national measurement networks, research or monitoring projects (cf. Supplementary Table S1). The long-term commitment and continuous work, including fieldwork, instrument maintenance and data curation by current and former principal investigators and the institutions providing financial support is greatly acknowledged and cannot be overestimated.

Permafrost data collection in Norway and Svalbard was primarily organized and funded through the Norwegian Meteorological Institute, the University of Oslo, the University Centre in Svalbard and the Svalbard Integrated Observing System (SIOS). In Switzerland, long-term permafrost data acquisition is organized by the Swiss Permafrost Monitoring Network PERMOS and financially supported by MeteoSwiss, in the framework of GCOS Switzerland, the Federal Office for the Environment FOEN and the Swiss Academy of Sciences SCNAT. Permafrost data collection in France is coordinated via PermaFrance, supported by Observatoire des Sciences de l'Univers de Grenoble (OSUG). Permafrost monitoring in the Italian Alps is maintained under an institutional service of the Regional Environmental Agency (ARPA Piemonte, ARPA Valle d'Aosta, ARPA Veneto), with the Stelvio borehole being managed by Insubria University with support from the Project PRIN 2015 # 2015N8F555 (MG). Data collection in Spain is supported by AGAUR ANTALP #2017-SGR-1102 (Catalonia, MO).

## Author contributions

The study was initiated by J.N. and K.I. following discussions on 20-year results of the European PACE transect (Etzelmüller et al.[7]) and as an in-depth study on warming in mountain permafrost that extends the global assessment by Biskaborn et al.[3]. J.N. performed the data collection, processing, and analyses and wrote the text with strong support from K.I., who is the principal co-author and performed the seasonal analyses. K.I., J.B., H.C., R.D., B.E., D.F., T.G., M.G., C.Ha., C.Hi., M.H., C.L., F.M., M.O., L.P., P.P., C.R., P.S., M.V., A.V. and M.P. are current principal investigators of at least one borehole site and contributed with long-term data acquisition and important expert knowledge on instrumentation and site characteristics. J.B., H.C., R.D., B.E., D.F., T.G., M.G., C.Ha., C.Hi., M.H., C.L., F.M., M.O., L.P., P.P., C.R., P.S., M.V., A.V. and M.P. contributed to the manuscript with critical feedback and discussions, which improved the analyses and interpretation.

## Competing interests

The authors declare no competing interests.
