## [Transparent Peer Review file · Nature Communications]

Enhanced warming of European mountain permafrost in the early 21st century

Corresponding Author: Dr Jeannette Noetzli

Version 0:

Reviewer comments:

Reviewer #1

(Remarks to the Author)

General comments to authors and editor

In the submitted manuscript by Noetzli et al. titled "Enhanced warming of European mountain permafrost in the early 21st century" the authors do a nice job of highlighting a rich and diverse dataset in multiple European regions to show that permafrost and depth temperatures have warmed in the early part of this century. The manuscript is a good study with a list of notable and quality authors well known in the permafrost community. Although the paper does not report on anything too surprising in many ways it articulates the impact and magnitudes of warming in the mountains of Europe very well. The collection of data and the way it is presented is a strength of this manuscript. Additionally, the manuscript is well written and easy to read. I feel that the manuscript is suitable for publication pending some minor revisions. I reviewed this manuscript with my PhD student and am happy to look at a revised version of the editors feel this is required. A list of more specific comments can be seen below.

Specific comments

Introduction –

Lines 82-109 - seems like a lengthy version of an abstract to me where they highlight their main findings. Is this out of place? Rather should they just make their objective clearer with a brief mention of their methodology?

Line 85 – Change "Altitudinal" to "Elevational" these are different things.

Line 99 – ground is misspelled.

Line 103-106 – Seems out of place in the introduction, definitely should be a discussion point though

Results

Section 2.1 had some very interesting findings. Technically this section has results in areas that do not have permafrost 10 m deep. Should the section heading be different in this case as it is not just permafrost temperatures you are looking at?

Line 127 – What is the significance of only having one borehole so far south in the Sierra Nevada mountains? Expand on this and why is this the case?

Line 131-132 – "Four time series from the Western and Central Alps just cover a decade." Something does not sound right with this sentence. Change 'just' to 'only'?

Line 135 – Why the melting point of ice rather than 0 °C? this is an odd way to state this, consider revising.

Line 136 – "high Arctic" should be written as "High Arctic".

Line 143 – Can you write the for example in short form? (e.g.).

Line 144-146 – Your use of low temperatures and low mountains in this sentence convolutes the message. Is there a way to reword this sentence?

Line 146-147 – What do you mean by ranges? Is this a trend or variability between sites?

Line 149-150 – Say "reached their maximum MAGT at 20 m depth" here as it is not entirely clear as is.

Line 152 – Reference for source listing extreme warmth in Svalbard in 2016 to 2018?

Section 2.2 – This section has some really interesting findings. I think they are presented well. My main note is how does removing the trends that are not statistically significant effect your results? Are some of the patterns you identify being at least in part shaped by non-significant results?

Line 174 – Are there any significant cooling trends (p-value <0.05)?

Section 2.2 – If you removed the non-significant trends how does this change the results? Do you see similar patterns in trends based on locations and substrate types?

Line 176 – Is there way to quantify and statistically test at how many of the sites the trends were lower and more robust at the 20 m depth?

Figure 5 – The 10 and 20y labels are a little hard to read as is. Green and red may be tricky for colour blind individuals, please consider revising.

Lines 236-239 – Very interesting result. Shows that local heterogeneity of ice content has more of an influence on warming trends of mountain permafrost than regional location.

Line 245-247 – Again, a very interesting result that ground temperatures outside of the ice bearing or rich permafrost follow more closely the trends in surface air temperature. One note here, do you need to define the acronym SAT? It is not written out before this.

Line 249 – Well done covering a variety of different combination of depths, time periods, and time spans.

Figure S5 – Why does this figure have a half eye showing the distribution of individual warming trends while your other figures have violins?

Section 2.4 – Clear section to me. Seasonal patterns of warming are discussed and result support a summer/autumn concentration of warming in the most recent decade while longer term it is based in winter.

Figure 7 – in the caption Colour is spelled “Color” when the UK/Can spelling is used before. Check the MS for consistency.

Line 295 to 297 – The final sentence of this paragraph should be revisited for clarity.

Line 299-301 – I think this is a good conclusion to make as their data is unique in including warmer ground temperature boreholes and cold permafrost boreholes in alpine regions.

Line 307-309 – Has warming been homogeneous in mountainous regions? Elevation dependent warming?

Line 313-314 – Have you given any examples of air temperature trends compared to ground temperature trends? Could you quantify the difference between these trends?

Line 316-317 – Would it be better to say “frozen and unfrozen water content” rather than ‘(un)frozen’?

352-355 – You use various/varies several times in this sentence. Look for possible synonyms.

370 – Spelling error, should be ‘unevenly’.

Methods

Line 434 – I checked your supplementary table (S1) and found that none of the boreholes were deployed later than 2013. Is it meant to be 2013 on this line rather than 2014?

4.3 Authors used the ordinary least squares method to calculate their moving 10- and 20-year trends. Alternatively, they tested the non-parametric method called the Theil-Sen Slope but found very little difference in the results. Should these be in the supplemental information? Authors also made the minimum data requirements 5 years for a decade and 10 years for 2 decades. Authors argue that inter seasonal and annual variability are not relevant to skewing the trends as the depths they are working at has very little signal from those. For SAT trends they took nearby air temperature stations at high elevations to reduce the effect of topography. Authors used the NORA3 reanalysis dataset to extend their Scandinavian and Svalbard air temperature datasets for trends. Please address these decisions in the revised MS.

Reviewer #2

(Remarks to the Author)

Reviewer #3

(Remarks to the Author)

The paper submitted by Noetzli et al. is a compilation of ground temperature records from boreholes located in European mountains. The records are overwhelmingly located in the European Alps but also include records from Sweden, Norway and Spain. The original component of the paper is to present a compilation that includes a much greater number of records from European mountains and to extend the record until the year 2022 (The records presented by Biskaborn et al. stopped in 2016). This dataset allows the authors to capture the great diversity of settings inherent to mountain topography and to discuss the role of ground ice in mitigating some of the temperature changes close to 0°C. They also discuss the “lag” inherent with ground temperature change in relation to surface temperature and its implications for future ground temperature warming.

In my view, this paper presents a unique set of ground temperature records that fully warrants its publication in Nature Communications. These datasets are collected over decades in difficult conditions and provide invaluable information and validation for engineers, modelers, and ultimately decision makers in mountain areas. These datasets are, however, seldom pulled together in synthesis papers and I applaud the authors for doing so. These papers have lasting impacts in the scientific community, as we have seen from the paper by Biskaborn et al. (2019).

I have suggested major revisions but the authors will appreciate that my concerns are mostly related to the quality of the writing and not to the quality of the scientific work. The manuscript is somewhat challenging to read for several reasons: 1. There is substantial redundancy that could be reduced 2. Some words are poorly chosen and could lead to misleading understanding of some of the processes being discussed 3. Some sections, particularly the ones on the history and necessity of observing networks, are superfluous or redundant with the conclusion.

I have provided detailed comments below on the manuscript and the figures.

Page 2, line 31: “Constituting”: Throughout the manuscript, the authors use present participles, often in convoluted sentences. I would strongly recommend to get rid of all present participles and simplify the sentences. I’ll therefore pass on the notes I made on the use of present participles in the manuscript (e.g. Page 2, lines 48 and 51), hoping that the authors can introduce these changes.

Page 2, line 32: “influences”. This is quite vague. “Impacts” might be a better option

Page 2, line 36: “and employing a more comprehensive data set”. I think I understand what the author mean, but it will be unclear to the reader why the data set is “more comprehensive data set” without mentioning Biskaborn et al. I would suggest

to remove these words

Page 2, line 39: "reduce". The authors used the past tense to report on the results in the abstract and now switch to present
Page 2, line 39: "obscure important transformations...." I don't think that it is possible to understand this sentence without having read the manuscript. The wording is convoluted and... ..obscure. I suggest to reword.

Page 2, line 44: "While the" This sentence is trying to say too many things at the same time. I would suggest to simplify and simply say that the warming of permafrost has not garnered the same attention.

Page 3, line 58: Is Permafrost temperature really the "primary" product of the ECV? I do not recall exactly the last iteration of the GTN-P or the ECV literature, but I do not remember it being made "primary" and Active layer depth and rockglacier kinematics secondary. De Facto it is, I agree with the authors, but since we are referring to the official networks here, we might as well be consistent with the official nomenclature.

Page 3, line 63 to line 69: This section is problematic in my view. It attempts to justify the originality of this paper compared to Biskaborn et al. 2019. The language, however, is very vague and convoluted: "were considered in a lump manner, etc.". I would recommend to focus on simple factual information: 1. There are more records than in Biskaborn et al. 2019 (give numbers) 2. They cover more settings 3. They extend the record by six more years. The text written here is not convincing. I also think you should clearly introduce the year 2022 here. I found the first mention of the last measurement year at page 9 (line 163)

Page 3, line 70 to 81: The history section is lengthy and I think quite unnecessary. To some extent, some of the points made in this section (e.g. "there is no specific Europe-wide coordination of permafrost data to date") should be addressed in the conclusion of the manuscript, but not to this section

Page 4, line 83: "distinct". Why say "distinct"? This is somewhat misleading and gives the feeling that is a dataset conceived "out of the system". I would suggest using "new". It is simpler and reflects the reality. This is a dataset with new records from boreholes that are not new.

Page 4, line 83: "next". This is also somewhat misleading. Intuitively, the reader is led to think that the permafrost areas were identified first, and that boreholes were then drilled "next" to those on purpose. One could introduce the term "recently thawed" to better describe the strategy.

Page 4, line 84: The dataset is now "new" and not "distinct"

Page 4, line 87: "Some of these...." This is very vague wording and there is no explanation of what is meant precisely with the "length" "enabling the evaluation". Since this is explained in the methods section, I would strongly advise to remove this sentence not to confuse the reader.

Page 4, line 88: Here, MAGT is introduced for the first time. This is reasonable, but until this section, the manuscript only introduced DZAA (Page 3, line 60). It is somewhat confusing to hear that MAGT will be used as the main indicator of change when DZAA was mentioned prominently earlier on. I would suggest adding a quick line after the introduction of MAGT to explain the main difference between MAGT and DZAA or to refer to section 4.3

Page 4, line 90: "running temperature trends". The term "running" disappears from the rest of the manuscript. Why using it here?

Page 4, line 94: "and on depth range". I am not exactly sure what the authors mean with this. Why not making another sentence to explain this precisely.

Page 4, line 95: "Warming rates are highest....": This is a sentence describing and explaining too many processes. The authors could possibly break it in two sentences

Page 4, line 103: "Permafrost warming..." This sentence is problematic. In the current context, it is true, but in absolute terms, it is not. I would suggest rewording.

Page 4, line 104: "Thermal disequilibrium". This is also a problematic term. I assume you mean that there is a "disequilibrium" between temperatures at the surface and temperatures at depth. It took me a while to understand what is meant and I started wondering what the equilibrium would be, if we are now in disequilibrium. I am not sure the use of disequilibrium helps here.

Page 5, line 109: "need to be endorsed". I would argue that these efforts are actually "endorsed" at the national and international levels. What is lacking is the capacity to perform all these tasks.

Figure 1: Very nice figure, it took me a while to find the Spanish borehole, but I assume it is difficult to improve the figure to address this.

Page 7, line 142: The order of the sites is a mystery to me. Alphabetical? By country? I am not sure this is consistent with the rest of the manuscript

Page 7, line 144: "permafrost temperatures in the Central....": The wording is awkward: "do not reach as low" "could be simplified to "are not as low". "Which is due to the lower mountains" is also awkward. "which is due to the lower elevations" would be more scientific.

Page 9, line 164: Here the change rate as two decimals (0.10), in most other places it has only one (0.1). Check through the manuscript.

Page 9, line 177: "more robust": Do you mean statistically? If yes, mention it

Page 9, line 184: "for the same group": I think you mean the same group of records as in Biskaborn et al. It is not obvious here. I would mention it directly

Page 10, line 190: Link to atmospheric conditions: It would help to see one example where atmospheric temperature, SAT and MAGT are shown on the same chart for a season or a decade to illustrate this statement.

Page 11, line 210. Caption Figure 4: typo with "calculated"

Page 11, line 213: The use of MGT to describe mean ground temperature is somewhat challenging to understand in the manuscript. The way I understand it is that it can use to describe means for ten or twenty years, but used primarily for ten year means. I do not understand why MAGT is used for annual means and not MDGT for decadal means. The rather loose use of MGT introduces some confusion. Later in the manuscript (Line 220), MGT is used single-handedly. I am left wondering if it is the decadal or 20-year mean, even though the authors mention the "corresponding decadal warming rates". It could be useful to introduce two acronyms for the decadal (MDGT) and the 20-year mean (MTGT).

Page 12, Figure 5: Minor comment: One could capitalize LOESS so that it clearly refers to the statistical method and not to

the sediment.

Page 12, line 241: "Picture". Too colloquial. Replace with "pattern"?

Page 13, line 246: As far as I could tell, this is the first time SAT is introduced, spell it out

Page 13, Figure 6: Inset (b): Why labelling it "no" when it was labelled "no pf" in Figure 4. "No pf" seems like a more understandable label

Page 13, Figure 6: The caption indicates that air temperature warming rates are shown in comparison to permafrost warming rates in (a). I looked at the chart several times, but could not see the air temperature warming rates. This is too bad because it would help to contextualize all the discussion points made about the lag between atmospheric warming and ground temperature warming (see my earlier comment about Page 10, line 190). I might not understand the chart well enough to see it though, but will probably not be the only one to do so.

Page 14: Generally, the paragraph from line 271 to 282 is very wordy and long.

Page 15, line 291: The term "Thermal monitoring" is awkward. It could be interpreted in many different ways. I would recommend changing it to "ground temperature monitoring".

Page 15, line 293 to line 295: The point made of snow here comes out of nowhere. It sounds quite specific and I do not understand why the authors make this point in the very first sentences of the discussion.

Page 15, figure 7: Small detail: The reader is referred to figure 6 for the box plot labelling, but in figure 6, the "number of time series" is labelled in black, whereas here it is in color.

Page 16, line 299: say "these studies" instead of "earlier studies" so that one does not have to cite them again here (since they are cited in the previous sentence).

Page 16, line 302: "only just reached lengths". Put the required length between brackets

Page 16, line 307: This refers to my previous comment on Figure 6: I don't see the SAT change on figure 6. Also, in figure 6, it say "air temperature" and here it says SAT. Is it the same? Are those used interchangeably? I would recommend only use one consistent wording. I

Page 16, line 308: "comparatively homogeneous": This is awkward wording. Say "consistent"?

Page 17, line 327: Give a number for the ground ice content, either gravimetric or volumetric

Page 17, line 353 "and contribute to understand". You mean "contribute to explain"?

Page 18, line 355: Here is the right location to discuss the role of snow

Page 18, line 356: "which": replace with "since it can temporarily..."

Page 18, line 370: Typo in unevenly

Page 18, line 374: "While a considerable number of time series". This sentence is very difficult to understand. Possibly break it in two sentences and simplify

Page 19, line 382: Here is the term robust is used in a non statistical sense, which strengthens my previous comment on the "statistically robust"

Page 19, line 386: Typo in maintenance

Page 19, line 386: Maintenance work WILL likely increase

Page 19, line 393: One could add an "s" to framework

Page 20, line 427, typo in agencies

Page 20, line 432: the borehole is in THE permafrost region

Page 24, line 519: "high elevation". Are these sites, or stations?

Reviewer #4

(Remarks to the Author)

The manuscript focused on the variability and trends in mountain permafrost temperature in Europe over the last thirty years. By compiling available observations on permafrost temperature from Arctic Svalbard to France, the authors demonstrate that permafrost in Europe is warming and especially so in the last decade. However, the rates of warming are not uniform as cold and ice-poor permafrost reacts on atmospheric warming stronger due to heat conduction, while ice-rich and warm permafrost show less warming due to latent heat effects associated with melting of ground ice.

These results are significant as have substantial implications on alpine ecosystems, infrastructure and slope hazards in the region. The results are generally consistent with other permafrost regions; however, it looks like that due to absence of strong buffers like thick snow and vegetation, mountain permafrost is stronger coupled to atmospheric climate and therefore less resilient compared to no-alpine permafrost. The authors show that the atmospheric signal will further propagate to the ground therefore affecting deeper permafrost with further consequences for slope stability and hydrology in the region.

I believe that this manuscript will be of great interest to diverse readership of the journal, as it provides (to my best knowledge) the first synthesis of permafrost data from boreholes in the entire Europe using uniform methodology. While the duration of records varies by region, in my opinion the methodology is explicit enough to address the sensors accuracy and drifts, record length, significance levels and use of OSL to demonstrate the individual trends. Then I am a bit concerned with aggregation of data, I guess geographic location and surface cover make sense. However, I find that classification of permafrost based on ice content and temperature somewhat arbitrary, so it would be great if authors clarify why warm vs cold permafrost at -2 C? Why not -3 C? Why ice bearing 10-50% vs ice poor (<10%) vs ice-rich (50-100%). Why not low ice content (0-10%), medium (10-20%), and high (>20%). To my knowledge these classes are not uniform across countries and construction manuals on permafrost so require some additional explanation or just a simple reference if available as the way you group your data may influence the results. Even from figure 5 it seems that rates can be quite variable both within and between proposed classes. For example, from figure 6c ice-poor and no-ice seem quite similar, does it mean that if permafrost has <10% of ice content than ice can be neglected as factor in warming trend? So ice becomes important at 10%? Does it really matter if it is 40 or 80% of ice or the same? Explaining a bit more on how/why you group classes will be

very helpful as it will not just report results by classes but help in process understanding and will be very valuable input to modeling communities. Similar with temperature is warm vs cold really matters? If it does matter where is the threshold (what happens at -2 C?). Interesting find that cold permafrost and no permafrost warm at the same rate based on Figure 6b, something that probably not the case in non-mountain permafrost regions where heat conduction of frozen vs non-frozen soils is quite different and maybe requires a bit of further explanation. Specifically, you stated "Differences in the observed warming patterns depend in the first place on the amount of latent heat exchanged in the ground, which is related to the temperature range and ground ice content, and on the depth" so can you elaborate more on this based on your observational data.

Overall, I found the text informative and easy to follow with figures very helpful addition to the text. I recommend to accept the manuscript pending clarifications (see above).

Version 1:

Reviewer comments:

Reviewer #1

(Remarks to the Author)

Review has satisfied my concerns and the MS looks good.

(Remarks on code availability)

Reviewer #2

(Remarks to the Author)

(Remarks on code availability)

Reviewer #3

(Remarks to the Author)

The authors have done a great job of reorganizing the paper and addressing my comments. I have no additional request.

(Remarks on code availability)

Reviewer #4

(Remarks to the Author)

The authors addressed comments and concerns raised by the reviewers and I believe that the revised manuscript improved significantly and is ready for the publication. I believe that this manuscript is a valuable contribution and will be received well by diverse readership of NComm.

Few minor suggestions:

L327 is awkward at the end and can it be rephrased without DZAA "and deeper permafrost that is not affected by seasonal variability" or something that is not too technical.

Table 1: Lat without long makes little sense, please add lon column to the table.

Figures are good quality. I suggest to change 0/12 to just 12 in figure 7 (right, both panels)

(Remarks on code availability)

Enhanced warming of European mountain permafrost in the early 21st century

Point-by-point reply to four reviewers' comments

We would like to thank the four reviewers for taking the time to provide thorough reviews with many helpful comments and suggestions, which have significantly helped to improve our manuscript. Our point-by-point reply is presented below. The reviewers' comments are printed in black, while our replies to them are in blue. Relevant text parts from the manuscript are given in *italic*.

Reviewer #1

General comments to authors and editor

In the submitted manuscript by Noetzli et al. titled "Enhanced warming of European mountain permafrost in the early 21st century" the authors do a nice job of highlighting a rich and diverse dataset in multiple European regions to show that permafrost and depth temperatures have warmed in the early part of this century. The manuscript is a good study with a list of notable and quality authors well known in the permafrost community. Although the paper does not report on anything too surprising in many ways it articulates the impact and magnitudes of warming in the mountains of Europe very well. The collection of data and the way it is presented is a strength of this manuscript. Additionally, the manuscript is well written and easy to read. I feel that the manuscript is suitable for publication pending some minor revisions. I reviewed this manuscript with my PhD student and am happy to look at a revised version of the editors feel this is required. A list of more specific comments can be seen below.

Reply to Reviewer #1

We thank Reviewer#1 very much for the positive and supportive comments and the thorough review, including the valuable input by a young researcher in the field (Reviewer #2). We are pleased that the value of the compiled data set of permafrost temperatures in European mountains was recognized. We addressed all comments raised by Reviewers #1 and 2 and adapted the manuscript accordingly (see detailed response below). We believe that the comments helped to considerably improve our manuscript and hope that the reviewers agree with our revisions. We have particularly reworked the introduction and discussion with some restructuring and moving of statements for a better focus on the scope and novelty of the study. To answer the comments by Reviewer #1, the influence of the non-significant warming rates and the use and comparison with surface air temperature are pointed out more clearly. We further included four additional time series obtained in mountain permafrost in Iceland, which complement the data set and the spatial representativeness of the study. Our detailed point-by-point reply is given below.

Specific comments

Where	Reviewer comment	Author reply
Introduction		
82-109	Seems like a lengthy version of an abstract to me where they highlight their main findings. Is this out of place? Rather should they just make their objective clearer with a brief mention of their methodology?	We thank the reviewer for pointing this out. We have revised the introduction, additionally addressing the comment regarding line 103–106 (below) and several related comments by another reviewer. We have shortened the text of the introduction (particularly the historical part) to more clearly present the novelty relative to

		earlier studies and our objective. In the last paragraph of the introduction we briefly summarize the key results and conclusions (according to the formatting instructions of Nature Communications https://www.nature.com/documents/ncomms-formatting-instructions.pdf)
85	Change “Altitudinal” to “Elevational” these are different things.	Corrected.
99	ground is misspelled.	Corrected.
103–106	Seems out of place in the introduction, definitely should be a discussion point though.	Yes, we agree with this comment. We moved the three statements of this paragraph to the discussion (cf. reply to the first comment above).
Results		
Section 2.1	Section 2.1 had some very interesting findings. Technically this section has results in areas that do not have permafrost 10 m deep. Should the section heading be different in this case as it is not just permafrost temperatures you are looking at?	Yes, this is a good point. We adapted the heading for a more general wording because also results from non-permafrost sites are presented in this section: Measured ground temperatures in European mountains
127	What is the significance of only having one borehole so far south in the Sierra Nevada mountains? Expand on this and why is this the case?	The borehole in the Sierra Nevada is the only borehole that exists in the Spanish mountains to date. This was again confirmed by our Spanish co-author. No permafrost borehole data are available from the Pyrenees. We aim to collect a data set that is as comprehensive as possible for European mountain regions. The borehole at Veleta Peak is considered an important complement as it is the only one in the region and it has been installed as part of a borehole transect in the framework of the EU-funded permafrost monitoring project PACE (cf., Harris et al. 2001). We clarify in the text: No data are available yet from other European mountain regions such as the Pyrenees.
131-132	“Four time series from the Western and Central Alps just cover a decade.” Something does not sound right with this sentence. Change ‘just’ to ‘only’?	Corrected as suggested, we changed just to only .
135	Why the melting point of ice rather than 0 °C? this is an odd way to state this, consider revising.	Yes, we agree and have adapted the sentence as suggested.
136	“high Arctic” should be written as “High Arctic”.	Corrected.
143	Can you write the for example in short form? (e.g.).	Corrected.
144-146	Your use of low temperatures and low mountains in this sentence convolutes the message. Is there a way to reword this sentence?	We reworded this and split it into two sentences to make it easier to read. Permafrost temperatures measured in the Central and Eastern Alps are not as low as in the Western Alps. This is due to the lower elevations in the former regions and the lack of observations >3000 m asl., and less due to climatic differences.
146-147	What do you mean by ranges? Is this a trend or variability between sites?	Here, we refer to the observed values of annual mean ground temperatures MGT_{1yr} at the non-permafrost sites. We replaced range up to with attain .

149-150	Say “reached their maximum MAGT at 20 m depth” here as it is not entirely clear as is.	We agree and have corrected the sentence as suggested.
152	Reference for source listing extreme warmth in Svalbard in 2016 to 2018?	We included a reference to Isaksen et al. (2022).
Section 2.2	This section has some really interesting findings. I think they are presented well. My main note is how does removing the trends that are not statistically significant effect your results? Are some of the patterns you identify being at least in part shaped by non-significant results? If you removed the non-significant trends how does this change the results? Do you see similar patterns in trends based on locations and substrate types?	We thank the reviewer for this positive comment and the relevant question on the influence of non-significant warming rates on the resulting pattern. To look at this question in detail, we calculated the mean values of all classes using only warming rates with $p\text{-values} \leq 0.05$. This results in a smaller n per calculated mean value of the distinguished classes, but the observed patterns remain the same. That is, they are not shaped by non-significant values: The number of warming rates with $p\text{-values} > 0.05$ is generally higher for shorter time periods (10 years) and for shallower depths (5 or 10 m). For 20-year time periods, $>80\%$ of the warming rates at 5 m depth have $p\text{-values} \leq 0.05$ and $>90\%$ for the lower depths. Removing non-significant values has hardly any influence on the pattern and a small effect on the presented mean warming rates ($< 0.1 \text{ } ^\circ\text{C dec}^{-1}$). For 10-year periods, the significant portion of warming rates is lower, particularly at 5 m depth (ca. 40%). We stated in our paper, that this depth is not suitable to derive decadal warming rates. If non-significant values are excluded, the mean warming rates of all time series is $0.17 \text{ } ^\circ\text{C dec}^{-1}$ higher at 10 m depth and $0.06 \text{ } ^\circ\text{C dec}^{-1}$ higher at 20 m depth. Results show higher mean values if warming rates with $p\text{-values} > 0.05$ are excluded particularly for low warming rates (i.e., for ice-bearing sites or at sites with temperatures close to $0 \text{ } ^\circ\text{C}$). This is caused by the fact that $p\text{-values}$ are generally higher for lower warming rates (see Figure A) => Excluding non-significant values would therefore lead to an overestimation of mean warming rates for the distinguished classes.  Figure A: Most recent warming rates of permafrost temperature time series plotted against the p-value of the linear regression. Each point is the warming rate of one time series with the three different depths distinguished by colour and the length of the period by the filling of the point: 10-year periods are shown with filled points and drawn lines, 20-year periods with circles and dashed lines. The statistical significance of the warming rates and its interpretation is discussed in the 5th paragraph of the discussion.

		Based on the considerations presented above, we added the following text to the end of the paragraph: Excluding warming rates with p-values > 0.05 would therefore not only reduce the number of observations, but result in an overestimation of mean warming rates, particularly for lower values. We also tested the resulting warming pattern based on warming rates with p-values < 0.05 only and found that they are not controlled by non-significant values. We also address the interpretation of the statistical significance in the methods part (this text was not adapted during revision): While the temporal aggregation has little effect on the obtained warming rates, it can reduce the statistical significance of the results, which should be considered when interpreting p-values. For a test, we additionally derived decadal warming rates based on monthly and daily time series for several locations with p-value > 0.05, which yielded significant p-values for most tested time series using monthly data, and for all using daily data.
174	Are there any significant cooling trends (p -value < 0.05)?	Only one of the four negative change rates for the most recent decade has a p-value < 0.05. This information was added to the text in brackets: Cooling rates were obtained for 4 locations in warm or near-zero permafrost in ice-bearing ground. They are, however, lower than $0.1 \text{ }^\circ\text{C dec}^{-1}$ (p-value < 0.05 only one time series).
176	Is there way to quantify and statistically test at how many of the sites the trends were lower and more robust at the 20 m depth?	We thank the reviewer for this interesting question and useful input. We have analysed this question for the time series for which recent warming rates are available for both, 10 m and 20 m depth. First, we calculated the difference of the warming rates at the two depths. A density plot shows that calculated differences are approximately symmetrical (see Figure B). Based on this, we performed a non-parametric Wilcoxon signed rank test on paired samples (i.e., warming rates at two different depths) to test if there is a significant difference.  Figure B: Density plot (red line) and histogram for the difference in most recent warming rates at 10 and 20 m depth. For 20-year periods, all warming rates considered have p-values ≤ 0.05 for both depths. The mean difference between 10 m and 20 m depth is $0.12 \text{ }^\circ\text{C dec}^{-1}$ ($0.41 \text{ vs. } 0.29 \text{ }^\circ\text{C dec}^{-1}$, $n=13$). The Wilcoxon signed rank test yields a p-value of 0.040, indicating significance. For 10-year periods, statistical robustness of the warming rates considered in the test is higher at 20 m depth than at 10 m depth (47% vs. 30%). The mean difference in warming rates is $0.13 \text{ }^\circ\text{C dec}^{-1}$ ($0.38 \text{ vs. } 0.25 \text{ }^\circ\text{C dec}^{-1}$, $n=35$). The Wilcoxon signed rank test provides a p-value of 0.007, indicating significance. This information was added to the paper at this place and as follows:

		A non-parametric Wilcoxon signed rank test on paired samples of warming rates at 10 m and 20 m depth shows a significant difference (0.05 level) for both 10-year (n=35, mean difference 0.13 °C dec⁻¹) and 20-year periods (n=13, mean difference 0.12 °C dec⁻¹).
Figure 5	The 10y and 20y labels are a little hard to read as is. Green and red may be tricky for colour blind individuals, please consider revising.	We thank the reviewer for noticing. For better readability, we changed the background of the labels to darker grey and printed the labels in bold with years written out. We further adapted the figure to avoid the combination of red and green. The red was changed to yellow and the red LOESS line to dark grey. Note: Figure 5 is Figure 6 in the revised manuscript. For consistency between figures, we adapted the colours, labels and background for all figures showing facets (incl. Supplementary Figures).
236-239	Very interesting result. Shows that local heterogeneity of ice content has more of an influence on warming trends of mountain permafrost than regional location.	This is well summarized.
245-247	Again, a very interesting result that ground temperatures outside of the ice bearing or rich permafrost follow more closely the trends in surface air temperature. One note here, do you need to define the acronym SAT? It is not written out before this.	We thank the reviewer for this comment and agree that this is a key result of our study. Following a comment by another reviewer, we added running warming rates of SAT to Figure 3 for comparison with ground temperature warming rates. SAT is now introduced at this place as it is the first occurrence of the acronym in the text.
249	Well done covering a variety of different combination of depths, time periods, and time spans.	We thank the reviewer for this comment!
Figure S5	Why does this figure have a half eye showing the distribution of individual warming trends while your other figures have violins?	We originally included the half eye in Supplementary Fig. 5 because of the smaller number of categories shown in the plot and to show more detail. But we agree, this is not necessary, and we changed the figure to be consistent with the other figures.
Section 2.4	Clear section to me. Seasonal patterns of warming are discussed and result support a summer/autumn concentration of warming in the most recent decade while longer term it is based in winter.	We thank the reviewer for this comment!
Figure 7	In the caption Colour is spelled “Color” when the UK/Can spelling is used before. Check the MS for consistency.	We checked the text and adapted the spelling to consistently follow the Oxford spelling.
Discussion		
295 to 297	The final sentence of this paragraph should be revisited for clarity.	We reformulated the final sentence of this paragraph, splitting it in two parts: Observed warming patterns depend mainly on the depth of observation and on the amount of latent heat exchanged in the ground. The latter depends on the ground temperature and the ground ice/water content.
299-301	I think this is a good conclusion to make as their data is unique in including warmer ground temperature boreholes and cold permafrost boreholes in alpine regions.	We thank the reviewer for this comment.

307-309	Has warming been homogeneous in mountainous regions? Elevation dependent warming?	Recent studies (e.g., Pepin et al. 2022; Beaumet et al. 2021) suggest that the occurrence of elevation dependent warming is contingent upon the period, season, or region being analysed. We acknowledge that the sentence was somewhat unclear and included updated references to cover more recent climate development: Since the 1980s, when the first mountain permafrost data became available, there was a marked increase in atmospheric warming in the Alps, which appears to have become more pronounced in recent years (Isotta et al. 2019). The highest rates of SAT increase are observed in summer, particularly at high elevations (>2000 m asl.) (Isotta et al. 2019; Beaumet et al. 2021).
313-314	Have you given any examples of air temperature trends compared to ground temperature trends? Could you quantify the difference between these trends?	SAT warming rates are presented in Figure 5a together with ground temperature warming rates (SAT warming rates are given for 30-year periods for the three regions Svalbard, Scandinavia, Alps). We additionally revised Figure 3 and added SAT warming rates for direct comparison with ground temperature warming rates. For a detailed qualitative comparison at the site scale, we lack long-term SAT data at the borehole locations for all sites. However, we compare the magnitude of the changes in the results section (Section on «Permafrost warming rates»). Warming rates in cold permafrost, at ice-poor bedrock sites and at non-permafrost sites are in a similar range and correspond to warming rates of surface air temperature (SAT) over the past 30 years (cf. methods for details on the calculation). Further, the third and fourth paragraph in the discussion section deals with the comparison and drivers of change rates in the ground and in the air.
316-317	Would it be better to say “frozen and unfrozen water content” rather than ‘(un)frozen’?	Yes, we agree. The sentence was corrected accordingly.
352-355	You use various/varies several times in this sentence. Look for possible synonyms.	The sentence was adapted to avoid the repetition of various/varies: The aforementioned effects influence the sites to varying degrees and contribute to explain the differences in warming rates observed across the surface covers and slope expositions of our monitoring sites.
370	Spelling error, should be ‘unevenly’	Corrected.
Methods		
434	I checked your supplementary table (S1) and found that none of the boreholes were deployed later than 2013. Is it meant to be 2013 on this line rather than 2014?	A minimum length of one decade is required to include a borehole time series in the study. Since we allow for one missing year at the beginning or end of the time series (cf. methods), a time series can start in 2014 to be considered. Therefore, 2014 is correct in the text. However, the borehole with the shortest time series included starts in 2013.
Section 4.3	Authors used the ordinary least squares method to calculate their moving 10- and 20-year trends. Alternatively, they tested the non-parametric method called the Theil-Sen Slope but found very little difference in the results. Should these be in the supplemental information?	We calculated all warming rates using both, the ordinary least squares (OLS) and the Theil-Sen Slope (TSS) methods. Because the differences of the resulting values between the two methods are marginal, we think that presenting them in the Supplementary Information does not add relevant content. We therefore suggest not to include it. To underline this, we calculated the mean and median difference of the warming rates based on the two methods and added this information in brackets at the respective place in the methods section:

		Comparing results from applying both methods, however, shows marginal differences (mean difference for all calculated warming rates <0.02 °C dec⁻¹, median difference <0.001 °C dec⁻¹).
Section 4.3	Authors also made the minimum data requirements 5 years for a decade and 10 years for 2 decades. Authors argue that inter seasonal and annual variability are not relevant to skewing the trends as the depths they are working at has very little signal from those.	Here, we are not sure we understand the reviewer's request, as no specific question or suggestion was formulated. We suspect the comment is triggered by the choices for data completeness made in our approach and thus take the opportunity to clarify them: The criteria for the minimum number of values to calculate a decadal warming rate follow those used by Biskaborn et al. (2019) and were adapted for 20-year periods. We argue that the following two points justify for the application of a linear regression on annual values: 1) annual values are aggregated data of continuous daily measurements, and 2) temperature variations are dampened with depth and influenced by longer time periods. Seasonal patterns are analysed and presented in the results section and discussed later in the text. The influence of inter-annual variability is shown in Figure 3 with the running warming rates.
Section 4.3	For SAT trends they took nearby air temperature stations at high elevations to reduce the effect of topography. Authors used the NORA3 reanalysis dataset to extend their Scandinavian and Svalbard air temperature datasets for trends. Please address these decisions in the revised MS.	SAT obtained at high elevation stations are highly correlated across the Alps. This correlation is much higher than with SAT from stations located nearby but in valley bottoms. The latter are influenced by local or topographic effects such as temperature inversions. In contrast, high elevation stations measure the temperature of the larger air mass and are hardly influenced by local effects. We added the example of temperature inversion as a local effect to give some more information in the text: To avoid effects of local topography or valleys (such as temperature inversions), we used data from three high elevation weather stations of the Federal Office for Meteorology and Climatology MeteoSwiss (...). Further, a sentence was included to provide clarification on the selection of the NORA3 data set: NORA3 was selected as it is currently the only high-resolution hindcast available to cover both the entire Scandinavian Peninsula and Svalbard up to 2023. Additionally, it was shown to outperform both the host reanalysis ERA5 and the earlier hydrostatic 10-km Norwegian Hindcast Archive (NORA10), especially in complex terrain (Haakenstad & Breivik 2022).

Reviewer #2

Reply to Reviewer #2

We thank Reviewer #2 very much for taking the time and providing useful comments, which helped to improve our work. We highly appreciate the initiative for and the input from Early Career Researchers. Our detailed replies to the comments raised by Reviewer #1 and Reviewer #2 are included above.

Reviewer #3

The paper submitted by Noetzli et al. is a compilation of ground temperature records from boreholes located in European mountains. The records are overwhelmingly located in the European Alps but also include records from Sweden, Norway and Spain. The original component of the paper is to present a compilation that includes a much greater number of records from European mountains and to extend the record until the year 2022 (The records presented by Biskaborn et al. stopped in 2016). This dataset allows the authors to capture the great diversity of settings inherent to mountain topography and to discuss the role of ground ice in mitigating some of the temperature changes close to 0°C. They also discuss the “lag” inherent with ground temperature change in relation to surface temperature and its implications for future ground temperature warming.

In my view, this paper presents a unique set of ground temperature records that fully warrants its publication in Nature Communications. These datasets are collected over decades in difficult conditions and provide invaluable information and validation for engineers, modelers, and ultimately decision makers in mountain areas. These datasets are, however, seldom pulled together in synthesis papers and I applaud the authors for doing so. These papers have lasting impacts in the scientific community, as we have seen from the paper by Biskaborn et al. (2019).

I have suggested major revisions, but the authors will appreciate that my concerns are mostly related to the quality of the writing and not to the quality of the scientific work. The manuscript is somewhat challenging to read for several reasons: 1. There is substantial redundancy that could be reduced 2. Some words are poorly chosen and could lead to misleading understanding of some of the processes being discussed 3. Some sections, particularly the ones on the history and necessity of observing networks, are superfluous or redundant with the conclusion.

I have provided detailed comments below on the manuscript and the figures.

Reply to Reviewer #3

We are grateful for the positive and constructive review and the many useful and detailed comments by Reviewer #3. They helped to considerably improve our manuscript and the readability of the text. We are pleased that Reviewer #3 emphasizes the importance and challenges of collecting long, consistent time series of permafrost data and acknowledges the effort to combine them in a synthesis paper on European mountains. We addressed all comments raised and adapted the manuscript accordingly. Our point-by-point reply to the detailed comments is given below. We answer the three main points mentioned as challenge for reading the manuscript as follows:

- 1) Redundancy: we checked the text for redundancy and removed text where it is not necessary. We shortened the introduction to avoid duplicate statements and particularly revised the results section on permafrost warming pattern: statements on the same pattern were combined and the part was structured following the different classifications. Figures 5 and 6 were swapped due to the changed order of reference in the text.
- 2) Choice of words: we adapted the manuscript according to the detailed comments, which improved the wording and clarity of the text. Additional comments on the wording by another reviewer also helped to improve readability.
- 3) Unnecessary sections: we reworked the introduction and particularly removed most of the historical paragraph. We included the points we think are most relevant for our study into other paragraphs of the introduction. The necessity of observing networks and their long-term support is important to mention in our view, but this now only appears in the discussion part. This reduces redundancy and makes for a more concise introduction.

Specific comments

Where	Reviewer #3 Comment	Author reply
Abstract		
Page 2, line 31	“Constituting”: Throughout the manuscript, the authors use present participles, often in convoluted sentences. I would strongly recommend to get rid of all present participles and simplify the sentences. I’ll therefore pass on the notes I made on the use of present participles in the manuscript (e.g. Page 2, lines 48 and 51), hoping that the authors can introduce these changes.	We checked the text for the use of present participles and simplified it where appropriate. Indeed, the readability and clearness of several sentences could be improved by reformulating them.
Page 2, line 32	“influences”. This is quite vague. “Impacts” might be a better option:	We changed the word as suggested.
Page 2, line 36	“and employing a more comprehensive data set”. I think I understand what the author mean, but it will be unclear to the reader why the data set is “more comprehensive data set” without mentioning Biskaborn et al. I would suggest to remove these words:“	We adapted the sentence to avoid using «more comprehensive»: ... because of accelerated warming and the use of a comprehensive data set.
Page 2, line 39	“reduce”. The authors used the past tense to report on the results in the abstract and now switch to present:	We agree on the use of the tenses and changed the abstract to present, except for where it concerns past events.
Page 2, line 39	“obscure important transformations...” I don’t think that it is possible to understand this sentence without having read the manuscript. The wording is convoluted and... ..obscure. I suggest to reword	We changed obscure to mask .
Introduction		
Page 2, line 44	“While the” This sentence is trying to say too many things at the same time. I would suggest to simplify and simply say that the warming of permafrost has not garnered the same attention	We agree that the start of the introduction was cumbersome to read and simplified the sentence as follows: While the retreat of glaciers in the 21st century has been widely recognised (Zemp et al. 2015; Hugonnet et al. 2021), permafrost changes in cold regions worldwide are far less visible despite being similarly important (Biskaborn et al. 2019; Smith et al. 2022; Isaksen, Lutz, et al. 2022; Romanovsky et al. 2010; Etzelmüller et al. 2020; Haberkorn et al. 2021; PERMOS 2023; Zhao et al. 2020; Magnin et al. 2023; Hock et al. 2019).
Page 3, line 58	Is Permafrost temperature really the “primary” product of the ECV? I do not recall exactly the last iteration of the GTN-P or the ECV literature, but I do not remember it being made “primary” and Active layer depth and rockglacier kinematics secondary. De Facto it is, I agree with the authors, but since we are referring to the official networks here, we might as well be consistent with the official nomenclature	We agree with this comment. There is no «official» ranking of the three products of the ECV permafrost. We adapted the text as follows: The assessment of permafrost changes relies on long-term records of ground temperatures measured in boreholes, which are the direct thermal observations defined as one of the products of the Essential Climate Variable (ECV) permafrost (Streletskiy et al. 2021; Noetzi et al. 2021; GCOS 2022).
Page 3, line 63 to line 69	This section is problematic in my view. It attempts to justify the originality of this paper compared to Biskaborn et al. 2019. The language, however, is very vague and convoluted: “were considered in a lump manner, etc.”. I would recommend to focus on simple factual information:	We reworked the introduction of the paper with particular attention to present the novelty and key results of the paper more clearly, to simplify sentences and to introduce the time-period considered.

	1. There are more records than in Biskaborn et al. 2019 (give numbers) 2. They cover more settings 3. They extend the record by six more years. The text written here is not convincing. I also think you should clearly introduce the year 2022 here. I found the first mention of the last measurement year at page 9 (line 163)	
Page 3, line 70 to 81	The history section is lengthy and I think quite unnecessary. To some extent, some of the points made in this section (e.g. “there is no specific Europe-wide coordination of permafrost data to date”) should be addressed in the conclusion of the manuscript, but not to this section	We included this paragraph to justify and describe our focus on European mountains. However, we agree that the section is somewhat long for an introduction and not all information is needed to follow our work. We therefore removed this section and included the information we consider most relevant into the other paragraphs of the introduction. However, we chose to keep the sentence that there was no organized European collection of permafrost data to date. This was a motivation for our work and is now an important result of it.
Page 4, line 83	“distinct”. Why say “distinct”? This is somewhat misleading and gives the feeling that is a dataset conceived “out of the system”. I would suggest using “new”. It is simpler and reflects the reality. This is a dataset with new records from boreholes that are not new	The formatting instructions of Nature Communications state that phrases like novel, new, unprecedented should be avoided. To follow this and at the same time make it clearer for the reader that the dataset was compiled particularly for this study, we adapted the sentence as follows (including a reference to the detailed description of the data collection in the methods section). We compiled a consistent data set of 64 ground temperature time series measured for at least one decade in or near permafrost areas. Data are obtained in boreholes extending at least 10 m in depth and the sites cover a wide range of landforms, elevations and latitudinal zones from high Arctic Svalbard and Scandinavia to Iceland, the Alps and the Sierra Nevada.
Page 4, line 83	“next”. This is also somewhat misleading. Intuitively, the reader is led to think that the permafrost areas were identified first, and that boreholes were then drilled “next” to those on purpose. One could introduce the term “recently thawed” to better describe the strategy.	Yes, the sentence was not clear. Next to was meant spatially, not temporally. We therefore changed it to near.
Page 4, line 84	The dataset is now “new” and not “distinct”	We deleted the word new at this place to avoid the use of new (cf. comment above on Page 4, Line 83).
Page 4, line 87	“Some of these....” This is very vague wording and there is no explanation of what is meant precisely with the “length” “enabling the evaluation”. Since this is explained in the methods section, I would strongly advise to remove this sentence not to confuse the reader.	We agree and have deleted this sentence.
Page 4, line 88	Here, MAGT is introduced for the first time. This is reasonable, but until this section, the manuscript only introduced DZAA (Page 3, line 60). It is somewhat confusing to hear that MAGT will be used as the main indicator of change when DZAA was mentioned prominently earlier on. I would suggest adding a quick line after the introduction of MAGT to explain the main difference between MAGT and DZAA or to refer to section 4.3:	The DZAA needs to be introduced above, because we refer to earlier studies reporting on warming rates for this depth. We follow the suggestion of the reviewer to refer at this place to the methods section for more explanations on the calculation of the trends.

Page 4, line 90	“running temperature trends”. The term “running” disappears from the rest of the manuscript. Why using it here?:	The term running is also used later in the text, mainly in the results section. We therefore prefer to keep it at this place. However, we changed it to running warming rates to be consistent with how the term is used further down in the text and to avoid confusion.
Page 4, line 94	“and on depth range”. I am not exactly sure what the authors mean with this. Why not making another sentence to explain this precisely.:	The differences in the warming rates mainly relate to two factors: 1) the amount of exchanged latent heat and 2) the considered depth. We changed the sentence and split it into two parts for more clarity: The warming patterns are primarily dependent on the depth of observation and the exchange of latent heat for phase change. In ice-bearing ground, the latter can significantly reduce warming rates when temperatures approach 0 °C thereby mask important changes such as increasing water content or ground ice melt.
Page 4, line 95	“Warming rates are highest...”: This is a sentence describing and explaining too many processes. The authors could possibly break it in two sentences	We agree and have made two sentences out of this rather complicated one: Warming rates are highest at cold permafrost sites with low ground ice content, where effects of latent heat are largely absent. Further, warming rates in the uppermost 10 meters are higher than at greater depths, where temperatures react with delay.
Page 4, line 103	“Permafrost warming...” This sentence is problematic. In the current context, it is true, but in absolute terms, it is not. I would suggest rewording.	We agree with the reviewer that also fast permafrost warming processes exists (such as for example via advective heat transport by water). Such processes are typically active in the uppermost metres. To be more general we changed the beginning of the sentence to Permafrost warming at depth ... Note that this paragraph was moved to the end of the discussion section to address another reviewer’s comment.
Page 4, line 104	“Thermal disequilibrium”. This is also a problematic term. I assume you mean that there is a “disequilibrium” between temperatures at the surface and temperatures at depth. It took me a while to understand what is meant and I started wondering what the equilibrium would be, if we are now in disequilibrium. I am not sure the use of disequilibrium helps here.	Yes, this is what we mean. Temperatures at depth are reflecting surface temperatures of past time periods. When the surface temperatures change fast, the differences to temperatures at depth become larger as the latter cannot adapt fast enough. This is what we mean when speaking of a thermal disequilibrium. We argue that this is an appropriate term to describe the current and particularly the future subsurface temperature field in permafrost regions. The term is also used in the context of inversion modelling in geothermal studies to reconstruct ground surface temperature histories. That said, we agree that a thermal equilibrium is a rather theoretical state. To clarify what we mean, we adapted the text as follows: The higher warming observed in the upper 10 metres of the ground compared to lower depths points to an increasing thermal disequilibrium between the uppermost metres in the permafrost and at depths below the DZAA. Note that this paragraph was moved to the end of the discussion section to address another reviewer’s comment.
Page 5, line 109	“need to be endorsed”. I would argue that these efforts are actually “endorsed” at the national and international levels. What is lacking is the capacity to perform all these tasks	We adapted endorsed to supported and conducted . The capacity to perform all the tasks is indeed what is mostly lacking, we agree. However, we believe that continued and further endorsement at all levels is important to increase this capacity.

Results		
Figure 1	Very nice figure, it took me a while to find the Spanish borehole, but I assume it is difficult to improve the figure to address this.	We agree that the light-yellow colour on white background of the Spanish borehole can be challenging to spot. We improved the visibility of the boreholes/ points with thicker contour lines. In addition, we added labels for the regions considered in the study. That way, the Sierra Nevada appears more prominently on the map.
Page 7, line 142	The order of the sites is a mystery to me. Alphabetical? By country? I am not sure this is consistent with the rest of the manuscript:	We assume the reviewer refers to the list of examples on line 143, which indeed did not follow a clear order. The sites are now ordered first by region (Alps, Scandinavia), then alphabetically, and with three examples for each region. Schafberg was mentioned twice by mistake. Permafrost with temperatures just close to 0 °C is found in all European mountain regions, mainly in rock glaciers and unconsolidated sediments at elevations around 2500 m asl. in the Alps, or in blockfields and debris at elevations around 1500 m asl. in Scandinavia (e.g., at Bellecombes, Gentianes, Les Attelas, Dovrefjell, Juvvasshoe, or Tron).
Page 7, line 144	“permafrost temperatures in the Central...”: The wording is awkward: “do not reach as low” could be simplified to “are not as low”. “Which is due to the lower mountains” is also awkward. “which is due to the lower elevations” would be more scientific.	The sentence was adapted for clarification: Permafrost temperatures measured in the Central and Eastern Alps are not as low as in the Western Alps. This is due to the lower elevations in the former regions and the lack of observations >3000 m asl., and less due to climatic differences.
Page 9, line 164	Here the change rate as two decimals (0.10), in most other places it has only one (0.1). Check through the manuscript	The manuscript was checked to present all change rates with two decimals throughout, unless the values are given as a comparison/order of magnitude such as «greater than 0.5 °C dec ⁻¹ ».
Page 9, line 177	“more robust”: Do you mean statistically? If yes, mention it	Yes, we mean robust in the sense of more statistically significant . The text was adapted accordingly.
Page 9, line 184	“for the same group”: I think you mean the same group of records as in Biskaborn et al. It is not obvious here. I would mention it directly	We mean the same group of time series as used to calculate the change rates for the decade 2007–2016 in our data set, which we compare to the values provided by Biskaborn et al. (2019). This is mentioned because our dataset includes additional time series to calculate the 2013–2022 warming rates (but they do not start early enough to calculate the 2007–2016 warming rate). We adapted the wording to make this clearer: Based on the time series in our dataset that cover the period 2007–2016, we obtain a mean warming rate of 0.39 °C dec⁻¹ at 10 m depth (n=26, 65% of p-values<0.05) and of 0.24 °C dec⁻¹ at 20 m depth (n=20, 85% of p-values<0.05) for 2007–2016, and slightly higher values of 0.40 and 0.26 °C dec⁻¹ for 10 and 20 m depth (64% and 84% of p-values<0.05) for 2013–2022, respectively.
Page 10, line 190	Link to atmospheric conditions: It would help to see one example where atmospheric temperature, SAT and MAGT are shown on the same chart for a season or a decade to illustrate this statement	Yes, we agree. SAT is shown together with permafrost temperature change rates later in the manuscript in Figure 5a. In the revised version, we added running warming rates of SAT also to Figure 3 for comparison with ground temperature change rates.
Page 11, line 210	Caption Figure 4: typo with “calculated”	Corrected.

Page 11, line 213	The use of MGT to describe mean ground temperature is somewhat challenging to understand in the manuscript. The way I understand it is that it can use to describe means for ten or twenty years, but used primarily for ten year means. I do not understand why MAGT is used for annual means and not MDGT for decadal means. The rather loose use of MGT introduces some confusion. Later in the manuscript (Line 220), MGT is used single-handedly. I am left wondering if it is the decadal or 20-year mean, even though the authors mention the “corresponding decadal warming rates”. It could be useful to introduce two acronyms for the decadal (MDGT) and the 20-year mean (MTGT)	We thank the reviewer for this comment. We agree that the use of the acronyms for annual or decadal aggregations can be confusing. Indeed, MAAT or MAGT are often used for mean values of longer periods than one year, such as for 30-year periods and climate normals. So we should choose different acronyms in our text. We changed the naming of the aggregated temperatures throughout the manuscript. We now use mean ground temperature (MGT) together with a subscript to indicate the period of averaging: MGT_{1yr}, MGT_{10yr}, MGT_{20yr}. We hope that way it is clear if we consider an annual mean value, a decadal or 20-year mean value.
Page 12, Figure 5	Minor comment: One could capitalize LOESS so that it clearly refers to the statistical method and not to the sediment.	The figure legend and caption were adapted accordingly. Note: Figure 5 is Figure 6 in the revised manuscript.
Page 12, line 241	“Picture”. Too colloquial. Replace with “pattern”?	Corrected.
Page 13, line 246	As far as I could tell, this is the first time SAT is introduced, spell it out	Correct. The acronym SAT is used the first time here and we therefore spell it out.
Page 13, Figure 6:	Inset (b): Why labelling it “no” when it was labelled “no pf” in Figure 4. “No pf” seems like a more understandable label	We adapted the label in Figure 5b to match the labelling of Figure 4 (please note that Figure 6 is Figure 5 in the revised version).
Page 13, Figure 6	The caption indicates that air temperature warming rates are shown in comparison to permafrost warming rates in (a). I looked at the chart several times, but could not see the air temperature warming rates. This is too bad because it would help to contextualize all the discussion points made about the lag between atmospheric warming and ground temperature warming (see my earlier comment about Page 10, line 190). I might not understand the chart well enough to see it though, but will probably not be the only one to do so.	Indeed, SAT change rates are shown for three regions in Figure 5a to the right in red colour (please note that Figure 6 is Figure 5 in the revised version). We renamed the labels to SAT for consistency and mentioned the SAT change more explicitly in the caption to help the reader see this information. Panel (a) shows recent decadal warming rates (2013–2022 or <4 years earlier) at 10 m depth for 4 major European mountain regions (Svalbard, Scandinavia North, Scandinavia South, Iceland, Eastern Alps and Western Alps) and compared to surface air temperature SAT warming rates for the 30-year period 1993–2022 in Svalbard (SAT Sv.), Scandinavia (SAT Sc.) and the European Alps (SAT Alps). In addition, SAT change rates are added to Figure 3 in the revised manuscript.
Page 14: 271 to 282	Generally, the paragraph from line 271 to 282 is very wordy and long.	We agree, the paragraph was too wordy and long. We reduced the length by about 50 words and adapted it as follows for better readability: Notable differences emerge between the most recent decade 2013–2022 and the 20-year period 2003–2022 (Fig. 7, left). For the latest decade, the highest monthly warming rates at 10 m depth originate from increased ground surface temperatures in summer and autumn, while for the 20-year period the highest monthly warming rates are primarily influenced by winter warming (Figure 7, right). For the 10-year period, summer and autumn contribute equally to warming for cold and warm permafrost, while for near-zero permafrost, autumn has a more significant impact. In areas without

		permafrost, summer is the dominant warming season. Over the 20-year period, winter emerges as the dominant warming season for cold-, warm and near-zero permafrost. Regionally, early winter appears as the peak warming time in Scandinavia and Svalbard, while late winter is most dominant in the Alps. Finally, non-permafrost areas experience the highest warming during summer and autumn.
Discussion		
Page 15, line 291	The term “Thermal monitoring” is awkward. It could be interpreted in many different ways. I would recommend changing it to “ground temperature monitoring”	The term was changed as suggested.
Page 15, line 293 to line 295	The point made of snow here comes out of nowhere. It sounds quite specific and I do not understand why the authors make this point in the very first sentences of the discussion.	We agree. We moved this point on snow to the snow-specific section later in the discussion (cf. comment related to Page 18, line 355 below).
Page 15, figure 7	Small detail: The reader is referred to figure 6 for the box plot labelling, but in figure 6, the “number of time series” is labelled in black, whereas here it is in color.:	We adapted the colours of the number of time series as well as the style and layout of Figure 7 to make it consistent with the other figures.
Page 16, line 299	say “these studies” instead of “earlier studies” so that one does not have to cite them again here (since they are cited in the previous sentence)	Corrected.
Page 16, line 302	“only just reached lengths”. Put the required length between brackets	We added a decade in brackets: Several additional boreholes have recently been drilled into high-elevation bedrock, but the related time series are yet too short (< 1 decade) to evaluate long-term changes. The information thus remains thus very scarce for the highest elevations (e.g. 4000 m asl. and higher in the Alps), where permafrost temperatures are likely a few degrees lower than those available to date.
Page 16, line 307	This refers to my previous comment on Figure 6: I don’t see the SAT change on figure 6. Also, in figure 6, it say “air temperature” and here it says SAT. Is it the same? Are those used interchangeably? I would recommend only use one consistent wording.	Surface air temperature (SAT) is shown for three regions (Svalbard, Scandinavia, Alps) in Figure 5a to the right in red colour (see also our reply to the comments on page 13 above; note that Figure 6 is Figure 5 in the revised manuscript). We adapted the labelling to SAT to make it clearer and mentioned this more explicitly in the caption (see reply to the earlier comment above). Yes, air temperature and SAT were used interchangeably in the submitted text. We have revised the manuscript to consistently use only one term (SAT) throughout the text.
Page 16, line 308	“comparatively homogeneous”: This is awkward wording. Say “consistent”?	We rewrote this sentence for a better wording and to consider another reviewer's comment: Since the 1980s, when the first permafrost data became available in the Alps, there was a marked atmospheric warming, which appears to have become more pronounced in recent years (Isotta et al. 2019). The highest rates of SAT increase are detected in summer, particularly in high elevations (>2000 masl) (Isotta et al. 2019; Beaumet et al. 2021).

Page 17, line 327	Give a number for the ground ice content, either gravimetric or volumetric	Information on ground ice content at the drill sites is typically available from semi-direct geophysical soundings or documentation during drilling. That is, precise quantitative information is hardly available, but a basic classification is possible based on this information. We assess in the applied classification if ground ice is present in rock fissures and pores only (as in bedrock), if there is considerable ground ice (as it is possible in unconsolidated sediments), or if excess ice and ice lenses are present (as in rock glaciers). This classification is qualitative, and therefore the thresholds given in the methods text are used to describe the categories rather than to classify the time series. We elaborate on this in the methods section and reworded this part to be clearer (see also reply to Reviewer #4). For the latter, usually only little qualitative information is available from drilling logs or semi-direct geophysical soundings. This means that only a basic classification on the amount of ground ice is possible. However, this is sufficient for a general assessment of the influence of latent heat exchange during warming. We therefore considered if ground ice is present in rock fissures and pores only as in bedrock, if there is considerable ground ice as it is possible in unconsolidated sediments, or if excess ice is present as in rock glaciers. We distinguished the four classes ...
Page 17, line 353	“and contribute to understand”. You mean “contribute to explain”?	Yes. The text was adapted accordingly.
Page 18, line 355	Here is the right location to discuss the role of snow:”	Yes, we agree. The previous text related to snow in the discussion was relocated and integrated into this section in the revised manuscript (cf. also comment on Page 15, line 293 to line 295 above).
Page 18, line 356	“which”: replace with “since it can temporarily... ”	Corrected.
Page 18, line 370	Typo in unevenly	Corrected.
Page 18, line 374	“While a considerable number of time series”. This sentence is very difficult to understand. Possibly break it in two sentences and simplify.	We agree, the sentence was too complex with too much content. We adapted the text for better readability: Several additional boreholes have recently been drilled into high-elevation bedrock, but the related time series are yet too short (< 1 decade) to evaluate long-term changes. The information thus remains very scarce for the highest elevations (e.g. 4000 m asl. and higher in the Alps), where permafrost temperatures are likely a few degrees lower than those available to date.
Page 19, line 382	Here is the term robust is used in a non statistical sense, which strengthens my previous comment on the “statistically robust”	Yes, you are right. Here the term robust it is not meant in a statistical sense. We use it to refer to the quality of the data.
Page 19, line 386	Typo in maintenance:	Corrected.

Page 19, line 386	Maintenance work WILL likely increase	Corrected.
Page 19, line 393	One could add an “s” to framework	We agree. s was added to framework.
Methods		
Page 20, line 427	typo in agencies	Corrected.
Page 20, line 432	the borehole is in THE permafrost region	We corrected the sentence to: (i) the borehole is located in permafrost.
Page 24, line 519	“high elevation”. Are these sites, or stations?	We mean weather stations. The word station was missing and is now included.

Reviewer #4

The manuscript focused on the variability and trends in mountain permafrost temperature in Europe over the last thirty years. By compiling available observations on permafrost temperature from Arctic Svalbard to France, the authors demonstrate that permafrost in Europe is warming and especially so in the last decade. However, the rates of warming are not uniform as cold and ice-poor permafrost reacts on atmospheric warming stronger due to heat conduction, while ice-rich and warm permafrost show less warming due to latent heat effects associated with melting of ground ice.

These results are significant as have substantial implications on alpine ecosystems, infrastructure and slope hazards in the region. The results are generally consistent with other permafrost regions; however, it looks like that due to absence of strong buffers like thick snow and vegetation, mountain permafrost is stronger coupled to atmospheric climate and therefore less resilient compared to no-alpine permafrost. The authors show that the atmospheric signal will further propagate to the ground therefore affecting deeper permafrost with further consequences for slope stability and hydrology in the region.

I believe that this manuscript will be of great interest to diverse readership of the journal, as it provides (to my best knowledge) the first synthesis of permafrost data from boreholes in the entire Europe using uniform methodology. While the duration of records varies by region, in my opinion the methodology is explicit enough to address the sensors accuracy and drifts, record length, significance levels and use of OSL to demonstrate the individual trends.

Then I am a bit concerned with aggregation of data, I guess geographic location and surface cover make sense. However, I find that classification of permafrost based on ice content and temperature somewhat arbitrary, so it would be great if authors clarify why warm vs cold permafrost at -2 C? Why not -3 C? Why ice bearing 10-50% vs ice poor (<10%) vs ice-rich (50-100%). Why not low ice content (0-10%), medium (10-20%), and high (>20%). To my knowledge these classes are not uniform across countries and construction manuals on permafrost so require some additional explanation or just a simple reference if available as the way you group your data may influence the results.

Even from figure 5 it seems that rates can be quite variable both within and between proposed classes. For example, from figure 6c ice-poor and no-ice seem quite similar, does it mean that if permafrost has <10% of ice content than ice can be neglected as factor in warming trend? So ice becomes important at 10%? Does it really matter if it is 40 or 80% of ice or the same?

Explaining a bit more on how/why you group classes will be very helpful as it will not just report results by classes but help in process understanding and will be very valuable input to modeling communities.

Similar with temperature is warm vs cold really matters? If it does matter where is the threshold (what happens at -2 C?). Interesting find that cold permafrost and no permafrost warm at the same rate based on Figure 6b, something that probably not the case in non-mountain permafrost regions where heat conduction of frozen vs non-frozen soils is quite different and maybe requires a bit of further explanation. Specifically, you stated "Differences in the observed warming patterns depend in the first place on the amount of latent heat exchanged in the ground, which is related to the temperature range and ground ice content, and on the depth" so can you elaborate more on this based on your observational data.

Overall, I found the text informative and easy to follow with figures very helpful addition to the text. I recommend to accept the manuscript pending clarifications (see above).

Reply to Reviewer #4

We thank Reviewer #4 very much for the positive and supporting evaluation of our manuscript. We'd like to take the opportunity to further explain here our classification regarding permafrost conditions (which is based on the temperature) and ground ice content (which is mainly based on the landform).

The main goal of the classification is to demonstrate that the variations in mountain permafrost warming rates follow a pattern, which is primarily driven by the exchange of latent heat during warming. The amount of latent heat exchanged depends on 1) how much ice/water is present in the ground and 2) the ground temperature.

Classification by ground ice content

Information on ground ice content at the drill sites is typically available from semi-direct geophysical soundings or observations during drilling. That is, precise quantitative information is hardly available, but a basic classification is possible based on this information. We assess in our approach if ground ice is present only in rock fissures and pores as in bedrock (ice-poor), if there is considerable ground ice as often found in permafrost in unconsolidated sediments (ice-bearing), or if excess ice is present as in rock glaciers (ice-rich). This classification is rather qualitative, and the thresholds indicated in the methods text are used to describe the categories rather than for the classification itself. Therefore, they have little effect on the resulting pattern of warming rates. The classification by ground ice content is considered constant for a site for our study and purpose.

The methods text was adapted to be clearer on this:

For the latter (i.e. ground ice content), usually only little qualitative information is available from drilling logs or semi-direct geophysical soundings. This means that only a basic classification on the amount of ground ice is possible. However, this is sufficient for a general assessment of the influence of latent heat exchange during warming. We therefore considered whether ground ice is only present in rock fissures and pores as is the case in bedrock, if there is considerable ground ice as is possible in unconsolidated sediments, or if excess ice is present, as in rock glaciers. We distinguished the four classes no ice (referring here to boreholes that are not in permafrost), ice-poor (ground ice only in pores or clefts; ground ice content up to around 10 Vol.%), ice-bearing (considerable and varying ground ice in unconsolidated sediments, ground ice content about 10–50 Vol.%), and ice-rich (ice supersaturation in rock glaciers; ground ice content up to 100 Vol. %).

Classification by permafrost conditions

The permafrost classes are assigned for each time series, depth and period based on the mean ground temperature of the 10-year (MGT_{10yr}) or 20-year period (MGT_{20yr}). The classification into cold and warm permafrost is widely used in the scientific literature. We use the threshold of $-2\text{ }^{\circ}\text{C}$, which is often used in the literature to distinguish between the two classes (e.g., Smith et al. 2022). In cold permafrost hardly any unfrozen water is available (Williams & Smith 1989). With increasing permafrost temperature the amount of unfrozen water exponentially increases towards $0\text{ }^{\circ}\text{C}$ (see for example Williams & Smith 1989; Lunardini 1991; Mottaghy & Rath 2006). To account for the fact that most of the phase transition takes place at ground temperatures just below $0\text{ }^{\circ}\text{C}$ we further distinguish between warm and near-zero permafrost. Here, latent heat effects are largest and temperature changes are very small. We chose $0.5\text{ }^{\circ}\text{C}$ as the threshold for «just below $0\text{ }^{\circ}\text{C}$ ». However, using a threshold $\pm 0.1\text{ }^{\circ}\text{C}$ has only little effect on the result (see Figure C).

Figure C: Histogram of mean ground temperatures MGT_{10yr} and the two depths 10 and 20 m for all time series and the most recent decade 2013–2022. The red lines indicate the thresholds used to define the permafrost classes, the numbers in rectangles are the number of time series in each class.

Finally, the distinction between near-zero and no permafrost based on $MGT_{10yr/20yr}$ and MGT_{1yr} was the most discussed among the authors. The key question was whether time series with positive maximum MGT_{1yr} during the period but with $MGT_{10yr/20yr} \leq 0\text{ }^{\circ}\text{C}$ should be considered «no» or «near zero» permafrost. Several approaches were tested and resulting pattern generally remained unchanged. Finally, we agreed to only consider time series where

MGT_{1y} does not exceed $0\text{ }^{\circ}\text{C}$ as near-zero permafrost. Time series with any $MGT_{1y} > 0\text{ }^{\circ}\text{C}$ during the period is classified as no permafrost because of the changing thermal processes when ground temperatures rise above $0\text{ }^{\circ}\text{C}$.

The relation between calculated warming rates and $MGT_{10yr/20yr}$ with the derived LOESS fit shown in Figure 6 confirms this classification. In this figure $MGT_{10yr/20yr}$ is shown as a continuous variable without classification into permafrost conditions. Smaller changes in the applied thresholds have only little effect on the resulting pattern and on the mean values of the permafrost condition classes.

We adapted the text in the methods section to better explain the classification by permafrost condition and the thresholds applied:

While surface cover and ground ice content were considered a constant characteristic for a borehole site, permafrost conditions were assigned for each period and depth. In addition to the classical distinction between warm and cold permafrost^t, we considered that the portion of unfrozen water, and hence the exchanged amount of latent heat, increases exponentially when the ground temperature increases towards $0\text{ }^{\circ}\text{C}$ e.g. ^{83,84}. Most of the phase change takes place at temperatures very close to $0\text{ }^{\circ}\text{C}$. We distinguished the following four classes: cold permafrost with a mean ground temperature for the 10-year or 20-year period (MGT_{10y} or MGT_{20y}) below $-2\text{ }^{\circ}\text{C}$ (negligible latent heat effects expected), warm permafrost with $MGT_{10y/20yr} \geq -2\text{ }^{\circ}\text{C}$ and $< -0.5\text{ }^{\circ}\text{C}$ (small latent heat effects, depending on ground freezing characteristics), and near-zero permafrost with $MGT_{10y/20yr} > -0.5\text{ }^{\circ}\text{C}$ and maximum $MGT_{1yr} \leq 0\text{ }^{\circ}\text{C}$ (considerable latent heat effects during phase change expected for locations containing substantial ground ice). We distinguished non-permafrost from near-zero permafrost when the maximum MGT_{1yr} is $> 0\text{ }^{\circ}\text{C}$. That is, time series for which permafrost was present at the beginning but not at the end of the considered period, were classified as no permafrost (and as thawed permafrost in Fig. 4).

References

- Beaumont, J., Ménégoz, M., Morin, S., Gallée, H., Fettweis, X., Six, D., Vincent, C., Wilhelm, B. & Anquetin, S. 2021. Twentieth century temperature and snow cover changes in the French Alps. *Regional Environmental Change* 21: 114. DOI: 10.1007/s10113-021-01830-x.
- Biskaborn, B.K., Smith, S.L., Noetzli, J., Matthes, H., Vieira, G., Streletskiy, D.A., Schoeneich, P., Romanovsky, V.E., Lewkowicz, T., Abramov, A., Allard, M., Boike, J., Cable, W.L., Christiansen, H.H., Delaloye, R., Diekmann, B., Drozdov, D., Etzelmüller, B., Grosse, G., Guglielmin, M., Ingeman-Nielsen, T., Isaksen, K., Ishikawa, M., Johansson, M., Johannsson, H., Joo, A., Kaverin, D., Kholodov, A., Konstantinov, P., Kroger, T., Lambiel, C., Lanckman, J.-P., Luo, D., Malkova, G., Meiklejohn, I., Moskalenko, N., Oliva, M., Phillips, M., Ramos, M., Sannel, A.B.K., Sergeev, D., Seybold, C., Skryabin, P., Vasiliev, A., Wu, Q., Yoshikawa, K., Zheleznyak, M. & Lantuit, H. 2019. Permafrost is warming at a global scale. *Nature Communications* 10: 264. DOI: 10.1038/s41467-018-08240-4.
- Etzelmüller, B., Guglielmin, M., Hauck, C., Hilbich, C., Hoelzle, M., Isaksen, K., Noetzli, J., Oliva, M. & Ramos, M. 2020. Twenty years of European mountain permafrost dynamics—the PACE legacy. *Environmental Research Letters* 15 : 104070. DOI: 10.1088/1748-9326/abae9d.
- GCOS 2022. *The 2022 GCOS Implementation Plan*, Geneva: World Meteorological Organization.
- Haakenstad, H. & Breivik, Ø. 2022. NORA3. Part II: Precipitation and Temperature Statistics in Complex Terrain Modeled with a Nonhydrostatic Model. *Journal of Applied Meteorology and Climatology* 61: 1549–1572. DOI: 10.1175/JAMC-D-22-0005.1.
- Haberkorn, A., Kenner, R., Noetzli, J. & Phillips, M. 2021. Changes in Ground Temperature and Dynamics in Mountain Permafrost in the Swiss Alps. *Frontiers in Earth Science* 9: 626686. DOI: 10.3389/feart.2021.626686.
- Harris, C., Haerberli, W., Vonder Mühl, D. & King, L. 2001. Permafrost monitoring in the high mountains of Europe: the PACE Project in its global context C. Harris, ed. *Permafrost and Periglacial Processes* 12: 3–11. DOI: 10.1002/ppp.377.
- Hock, R., Alder, C., Caceres, B., Gruber, S., Hirabayashi, Y., Jackson, M., Kääh, A., Kang, S., Kutuzov, S., Milner, A., Molau, U., Morin, S., Orlove, B. & Stelzer, H. 2019. High mountain areas. In *IPCC Special Report on the Ocean and Cryosphere in a Changing Climate*. pp. 131–202.
- Hugonnet, R., McNabb, R., Berthier, E., Menounos, B., Nuth, C., Girod, L., Farinotti, D., Huss, M., Dussailant, I., Brun, F. & Kääh, A. 2021. Accelerated global glacier mass loss in the early twenty-first century. *Nature* 592: 726–731. DOI: 10.1038/s41586-021-03436-z.
- Isaksen, K., Lutz, J., Sørensen, A.M., Godøy, Ø., Ferrighi, L., Eastwood, S. & Aaboe, S. 2022. Advances in operational permafrost monitoring on Svalbard and in Norway. *Environmental Research Letters* 17: 095012. DOI: 10.1088/1748-9326/ac8e1c.

- Isaksen, K., Nordli, Ø., Ivanov, B., Køltzow, M.A.Ø., Aaboe, S., Gjeltén, H.M., Mezghani, A., Eastwood, S., Førland, E., Benestad, R.E., Hanssen-Bauer, I., Brækkan, R., Sviashchennikov, P., Demin, V., Revina, A. & Karandasheva, T. 2022. Exceptional warming over the Barents area. *Scientific Reports* 12: 9371. DOI: 10.1038/s41598-022-13568-5
- Isotta, F.A., Begert, M. & Frei, C. 2019. Long-Term Consistent Monthly Temperature and Precipitation Grid Data Sets for Switzerland Over the Past 150 Years. *Journal of Geophysical Research: Atmospheres* 124: 3783–3799. DOI: 10.1029/2018JD029910.
- Lunardini, V.J. 1991. *Heat transfer with freezing and thawing*. Developments in Geotechnical Engineering, CRREL, U.S. Army Corps of Engineers, Hanover, NH, USA.
- Magnin, F., Ravel, L., Bodin, X., Deline, P., Malet, E., Krysiński, J. & Schoeneich, P. 2023. Main results of permafrost monitoring in the French Alps through the *PermaFrance* network over the period 2010–2022. *Permafrost and Periglacial Processes* ppp.2209. DOI: 10.1002/ppp.2209.
- Mottaghy, D. & Rath, V. 2006. Latent heat effects in subsurface heat transport modelling and their impact on palaeotemperature reconstructions. *Geophysical Journal International* 164: 236–245. DOI: 10.1111/j.1365-246x.2005.02843.x.
- Noetzli, J., Aronson, L.U., Bast, A., Beutel, J., Delaloye, R., Farinotti, D., Gruber, S., Gubler, H., Haeblerli, W., Hasler, A., Hauck, C., Hiller, M., Hoelzle, M., Lambiel, C., Pellet, C., Springman, S.M., Vonder Muehll, D. & Phillips, M. 2021. Best Practice for Measuring Permafrost Temperature in Boreholes Based on the Experience in the Swiss Alps. *Frontiers in Earth Science* 9: 607875. DOI: 10.3389/feart.2021.607875.
- Pepin, N.C., Arnone, E., Gobiet, A., Haslinger, K., Kotlarski, S., Notarnicola, C., Palazzi, E., Seibert, P., Serafin, S., Schöner, W., Terzagio, S., Thornton, J.M., Vuille, M. & Adler, C. 2022. Climate Changes and Their Elevational Patterns in the Mountains of the World. *Reviews of Geophysics* 60: DOI: 10.1029/2020RG000730.
- PERMOS 2023. *Swiss Permafrost Bulletin 2022*, Fribourg and Davos: Swiss Permafrost Monitoring Network PERMOS.
- Romanovsky, V.E., Smith, S.L. & Christiansen, H.H. 2010. Permafrost thermal state in the polar Northern Hemisphere during the international polar year 2007–2009: a synthesis. *Permafrost and Periglacial Processes* 21 : 106–116. DOI: 10.1002/ppp.689.
- Smith, S.L., O’Neill, H.B., Isaksen, K., Noetzli, J. & Romanovsky, V.E. 2022. The changing thermal state of permafrost. *Nature Reviews Earth and Environment* 3: 10–23. DOI: doi.org/10.1038/s43017-021-00240-1.
- Streletskiy, D., Noetzli, J., Smith, S.L., Vieira, G., Schoeneich, P., Hrbacek, F. & Irrgang, A.M. 2021. Strategy and Implementation Plan for the Global Terrestrial Network for Permafrost (GTN-P) 2021–2024. DOI: 10.5281/ZENODO.6075468.
- Williams, P.J. & Smith, M.W. 1989. *The Frozen Earth*, Cambridge: Cambridge University Press.
- Zemp, M., Frey, H., Gaertner-Roer, I., Nussbaumer, S.U., Hoelzle, M., Paul, F., Haeblerli, W., Denzinger, F., Ahlstrom, A.P., Anderson, B., Bajracharya, S., Baroni, C., Braun, L.N., Caceres, B.E., Casassa, G., Cobos, G., Davila, L.R., Delgado Granados, H., Demuth, M.N., Espizua, L., Fischer, A., Fujita, K., Gadek, B., Ghazanfar, A., Hagen, J.O., Holmlund, P., Karimi, N., Li, Z., Pelto, M., Pitte, P., Popovnin, V.V., Portocarrero, C.A., Prinz, R., Sangewar, C.V., Severskiy, I., Sigurosson, O., Soruco, A., Usabaliyev, R., Vincent, C. & Correspondents, W.N. 2015. Historically unprecedented global glacier decline in the early 21st century. *Journal of Glaciology* 61: 745+. DOI: 10.3189/2015JoG15J017
- Zhao, L., Zou, D., Hu, G., Du, E., Pang, Q., Xiao, Y., Li, R., Sheng, Y., Wu, X., Sun, Z., Wang, L., Wang, C., Ma, L., Zhou, H. & Liu, S. 2020. Changing climate and the permafrost environment on the Qinghai–Tibet (Xizang) plateau. *Permafrost and Periglacial Processes* 31: 396–405. DOI: 10.1002/ppp.2056

Enhanced warming of European mountain permafrost in the early 21st century

Point-by-point reply to final comments by reviewers

We would like to thank the four reviewers for taking the time to assess our revision and replies. We are pleased that we were able to answer the comments and questions satisfactorily.

Here, we provide the reply to the remaining comments by Reviewer #4, which we addressed with the final revision. The reviewers' comments are printed in black, while our replies to them are in blue.

Reviewer #4

The authors addressed comments and concerns raised by the reviewers and I believe that the revised manuscript improved significantly and is ready for the publication. I believe that this manuscript is a valuable contribution and will be received well by diverse readership of NComm.

Few minor suggestions:

L327 is awkward at the end and can it be rephrased without DZAA “and deeper permafrost that is not affected by seasonal variability” or something that is not too technical.

We agree with this comment. We have rephrased the sentence as follows:

The higher warming observed in the upper 10 metres of the ground compared to 20 m depth points to an increasing thermal disequilibrium between the uppermost metres and permafrost at depth.

Table 1: Lat without long makes little sense, please add lon column to the table.

Yes, you are right. To avoid making the large table even larger, we have inserted the region of the borehole site instead of the coordinates. The coordinates (Lat/Lon) can be found in the Supplementary Table 1.

Figures are good quality. I suggest to change 0/12 to just 12 in figure 7 (right, both panels)

We have adapted Figure 7 accordingly.